# Par3 cooperates with Sanpodo for the assembly of Notch clusters following asymmetric division of *Drosophila* sensory organ precursor cells

Elise Houssin[1,2†], Mathieu Pinot[1,2†], Karen Bellec[1,2‡], Roland Le Borgne[1,2*]

[1]Univ Rennes, CNRS, IGDR (Institut de Génétique et Développement de Rennes) - UMR 6290, F- 35000, Rennes, France; [2]Equipe Labellisée Ligue Nationale contre le cancer, Glasgow, United Kingdom

**\*For correspondence:**
roland.leborgne@univ-rennes1.fr

[†]These authors contributed equally to this work

**Present address:** [‡]Wolfson Wohl Cancer Research Centre, Institute of Cancer Sciences, University of Glasgow, Glasgow, United Kingdom

**Competing interest:** The authors declare that no competing interests exist.

**Abstract** In multiple cell lineages, Delta-Notch signalling regulates cell fate decisions owing to unidirectional signalling between daughter cells. In *Drosophila* pupal sensory organ lineage, Notch regulates the intra-lineage pIIa/pIIb fate decision at cytokinesis. Notch and Delta that localise apically and basally at the pIIa-pIIb interface are expressed at low levels and their residence time at the plasma membrane is in the order of minutes. How Delta can effectively interact with Notch to trigger signalling from a large plasma membrane area remains poorly understood. Here, we report that the signalling interface possesses a unique apico-basal polarity with Par3/Bazooka localising in the form of nano-clusters at the apical and basal level. Notch is preferentially targeted to the pIIa-pIIb interface, where it co-clusters with Bazooka and its cofactor Sanpodo. Clusters whose assembly relies on Bazooka and Sanpodo activities are also positive for Neuralized, the E3 ligase required for Delta activity. We propose that the nano-clusters act as snap buttons at the new pIIa-pIIb interface to allow efficient intra-lineage signalling.

## Introduction

Notch is the receptor of an evolutionarily conserved cell-cell signalling pathway that controls fate acquisition in numerous processes throughout metazoan development (*Kopan and Ilagan, 2009*). Within many cell lineages, following the division of a precursor cell, Notch activation regulates binary fate choice between daughter cells (*Bertet et al., 2014*; *Bivik et al., 2016*; *Dong et al., 2012*; *Ohlstein and Spradling, 2007*; *Pardo-Saganta et al., 2015*; *San-Juan and Baonza, 2011*). In the majority of cases, the Notch receptor is activated by transmembrane ligands present in the adjacent cell. Following binding to Notch, endocytosis of the ligand generates pulling forces, driving a change in the conformation of the Notch extracellular domain, leading to the exposure of the S2 cleavage site of Notch (*Gordon et al., 2015*; *Langridge and Struhl, 2017*; *Meloty-Kapella et al., 2012*; *Seo et al., 2016*; *Shergill et al., 2012*; *Wang and Ha, 2013*). This mechanosensitive cleavage is followed by a constitutive proteolytic cleavage of Notch by the gamma secretase complex (*Mumm et al., 2000*; *Struhl and Adachi, 2000*). This gives rise to the transcriptionally active Notch intracellular domain (NICD) (*Kopan and Ilagan, 2009*). Since proteolytic activation of the Notch receptor is irreversible, Notch activation needs to be tightly controlled in time and in space. However, the spatio-temporal cascade of the events remains poorly characterised.

Sensory organ precursors (SOPs) of the pupal notum of *Drosophila* have been instrumental in the study of intra-lineage, Notch-dependent fate decisions (*Schweisguth, 2015*). SOPs are epithelial cells that divide asymmetrically within the plane of a single-layer epithelium to generate two

daughter cells, an anterior pIIb cell and a posterior pIIa cell, which are precursors of internal and external cells of the sensory organ, respectively. The pIIa-pIIb fate acquisition relies on the differential activation of Notch during cytokinesis as a result of the unequal partitioning of Numb and Neuralized (Neur) in the pIIb cell (*Le Borgne and Schweisguth, 2003*; *Rhyu et al., 1994*). In pIIb, Numb interacts with and regulates the trafficking of Sanpodo (Spdo), a four-pass transmembrane protein interacting with Notch and required for Notch signalling (*Babaoglan et al., 2009*; *Cotton et al., 2013*; *Couturier et al., 2013*; *O'Connor-Giles and Skeath, 2003*, *Johnson et al., 2016*; *Upadhyay et al., 2013*). Numb causes the targeting of Spdo/Notch to late endosomes (*Cotton et al., 2013*; *Couturier et al., 2013*), whereas Neur promotes the endocytosis of the Notch ligand Delta (Dl) (*Le Borgne and Schweisguth, 2003*), so that Notch is inhibited in pIIb and activated in pIIa. During SOP cytokinesis, two pools of Notch, located apically and basally to the midbody, are present at the pIIb-pIIa interface and both contribute to Notch signalling (*Bellec et al., 2021*; *Trylinski et al., 2017*). Previous studies based on photobleaching and photoconversion experiments have revealed that the basal pool of Notch is the main contributor of NICD (*Trylinski et al., 2017*). However, it remains largely unknown how the two pools of Notch are targeted along the pIIa-pIIb interface to promote this private intra-lineage cell-cell communication rather than with the neighbouring epidermal cells.

In vertebrates, the scaffolding protein Par3 regulates Numb-mediated trafficking of integrin and amyloid precursor protein (APP). Indeed, by binding to the phosphotyrosine domain of Numb, Par3 precludes Numb from binding to integrin, thus hindering Numb from causing integrin endocytosis (*Nishimura and Kaibuchi, 2007*). Similarly, Par3 interferes with the interaction between Numb and APP. In the absence of Par3, there is an increase in Numb-APP interaction leading to decreased surface APP and increased targeting of APP to late endosomal-lysosomal compartments (*Sun et al., 2016*). Whether Numb and Bazooka (Baz), the *Drosophila* ortholog of Par3, interfere with each other to control Spdo/Notch trafficking in SOP daughters is unknown.

In addition to regulating Numb-mediated membrane trafficking, Par3 regulates adherens junction (AJs) organisation and forms a complex with Par6 and atypical protein kinase C (aPKC), a complex that is essential in the establishment or maintenance of epithelial cell apico-basal polarity (*Assemat et al., 2008*; *Goldstein and Macara, 2007*; *Laprise and Tepass, 2011*; *Nelson, 2003*; *Rodriguez-Boulan and Macara, 2014*; *St Johnston and Ahringer, 2010*). During SOP mitosis, the unequal segregation of Numb relies on the SOP-specific remodelling of polarity modules and the phosphorylation by the Baz-aPKC-Par6 complex (*Bellaïche et al., 2001*; *Wirtz-Peitz et al., 2008*). Prior to mitotic entry, aPKC-Par6 are in complex with the tumour suppressor lethal giant larvae (Lgl). Assembly of the Baz-aPKC-Par6 complex is initiated upon phosphorylation of Par6 by the mitotic kinase AuroraA (AurA), then causing the autoactivation of aPKC. aPKC next triggers the phosphorylation of Lgl. Phosphorylated aPKC and Par6 can assemble with Baz (*Wirtz-Peitz et al., 2008*).

The Baz complex localises at the posterior apical and lateral cortex, while discs-large (Dlg) and partner of inscuteable (Pins) accumulate at the anterior lateral cortex during SOP mitosis (*Bellaïche et al., 2001*; *Roegiers et al., 2001*). Baz complex phosphorylates Numb at the posterior cortex, thereby preventing Numb to localise there, resulting in the unequal distribution of Numb in the anterior cortex (*Smith et al., 2007*; *Wirtz-Peitz et al., 2008*). Following degradation of AurA at metaphase to anaphase transition, Baz may be released from the Par6-aPKC complex. The localisation and potential functions of Baz versus the aPKC/Par6 complex during cytokinesis of SOP, as well as the consequence of the polarity remodelling at mitosis on the apico-basal polarity of the pIIa-pIIb interface at the time of Notch activation, are unknown.

In this study, we analyse the remodelling of cell-cell junction markers and polarity determinants throughout the process of SOP cytokinesis and compare it to that of epidermal cell cytokinesis. We report that, in the SOP, the PAR complex is dismantled during cytokinesis with aPKC redistributing in intracellular apical compartments, while Baz localises into apical and lateral clusters along the pIIa-pIIb interface together with Notch and Spdo. Analyses of clone borders reveal that Notch is not uniformly distributed at the plasma membrane but is instead selectively enriched at the pIIa-pIIb interface, indicative of a polarised transport mechanism towards the SOP daughter cells interface. Baz and Spdo, but not Notch, are required for the formation of the clusters. Neur localises in the clusters while Numb prevents cluster occurrence. We propose a model in which Baz, Notch, and Spdo co-cluster to favour signalling.

## Results

### Atypical apico-basal polarity of the pIIa-pIIb interface following SOP division

As SOP undergoes a specific redistribution of polarity modules during division (*Bellaïche et al., 2001*), we started by investigating the remodelling of junctional complexes during cytokinesis and the resulting apico-basal polarity of the nascent pIIa-pIIb cell interface from which Notch is activated. We previously reported that formation of the novel adhesive pIIa-pIIb interface, visualised with DE-Cadherin-GFP (E-Cad), is assembled with similar kinetics to those of epidermal daughters (*Founounou et al., 2013*). Here, we live-monitored and compared the localisation of septate junction (SJ) markers during SOP versus epidermal cell cytokinesis. All fluorescent markers are inserted at the locus, giving rise to functional reporters expressed at physiological level.

SOPs and daughters were identified using the nuclear marker Histone 2B-IRFP (H2B-IR) or the plasma membrane marker growth-associated protein 43 (GAP43)-IRFP, expressed under the SOP-specific *neur* minimal promoter. GAP43 consists in the first 20 amino acids of GAP43, which contains a dual palmitoylation signal that tightly anchors the protein to the inner leaflet of the plasma membrane. The progression of mitosis was tracked by cell shape. Cells in prometaphase are spherical, and the metaphase to anaphase transition is determined by the moment when the sphericity of the cells is lost, prior to adoption of a peanut shape. In every case, the transition from metaphase to anaphase was considered to be t0 (with time indicated in min:s). Dlg-GFP (*Woods and Bryant, 1991*) and neuroglian-YFP (Nrg-YFP; *Genova and Fehon, 2003*), two markers of SJs, are progressively recruited at the new pIIa-pIIb interface, immediately basal to the AJs around 25 min after anaphase, with similar kinetics as in epidermal daughters (*Figure 1—figure supplement 1A–B' and E*), indicating that SJ assembly occurs with similar kinetics along the new pIIa-pIIb and epidermal cell interfaces.

We next analysed the localisation of the component of the subapical complex Crumbs (Crb) as well as two members of the Par complex: aPKC and Baz. Crb-GFP is detected faster at the new apical pIIa-pIIb interface than between epidermal daughters (*Figure 1A–A"*, *Figure 1—figure supplement 1C and E*). Then, while Crb remains localised at the apical interface of epidermal cells (*Figure 1—figure supplement 1C–C'*), in pIIa and pIIb cells Crb-GFP localises primarily in apical cytoplasmic puncta (t13 ±4 min; *Figure 1A–A"* and *Figure 1—figure supplement 1C'*), at the expense of the pIIa-pIIb cell interface (t23 ±4 min). A similar behaviour was observed for aPKC-GFP, which is first localised at the new pIIa-pIIb interface (t7 ± 1 min; *Figure 1B–B"*, and *Figure 1—figure supplement 1D–E*) and then redistributed in part to cytoplasmic puncta primarily in the pIIa cell. In striking contrast to aPKC and Crb, Baz-GFP is not relocalised in apical cytoplasmic puncta. Instead, Baz-GFP is localised both at the pIIa-pIIb interface and is also enriched at the posterior pole of the pIIa cell (*Figure 1C–C"* and t9 min), in agreement with previous reports (*Le Borgne et al., 2002*; *Roegiers et al., 2001*). In comparison, at the epidermal daughter cell interface, there is no enrichment of Baz-GFP (*Figure 1—figure supplement 1F*, t12 min, upper panels). Finally, Baz-GFP is localised in punctate structures at the lateral pIIa-pIIb interface (*Figure 1C–C"* middle panels and orthogonal views, see also *Video 1*). These punctate structures, which we will refer to as lateral interface clusters and are specific to the interface of SOP daughter cells, appear at the same time as the first apical Baz clusters, ~ 10 min after the onset of anaphase (*Figure 1C–C"*).

Based on the distribution of Crb, aPKC, and Baz, we propose that apico-basal polarity is remodelled during SOP cytokinesis giving rise to a pIIa-pIIb interface with an atypical polarity. We next investigated the possible role of this polarity reshaping on Notch receptor localisation and activation.

### Polarity remodelling of the SOP is concomitant with localisation of Notch at the pIIa-pIIb interface

The localisation of Baz is reminiscent of that reported for Notch::GFP[CRISPR] (*Bellec et al., 2018*), a GFP-tagged version of Notch thereafter referred to as NiGFP. NiGFP transiently distributes at the apical and lateral pIIa-pIIb interface (*Figure 1D–D"*) prior to its proteolytic activation and subsequent targeting into the nucleus of the pIIa cell (*Bellec et al., 2018*; *Bellec et al., 2021*; *Couturier et al., 2012*; *Trylinski et al., 2017*). Like Baz, Notch is detected in punctate structures at the lateral interface of the pIIa-pIIb cells but not that of epidermal daughter cells (*Figure 1D–D"* and *Figure 1—figure supplement 1G*).

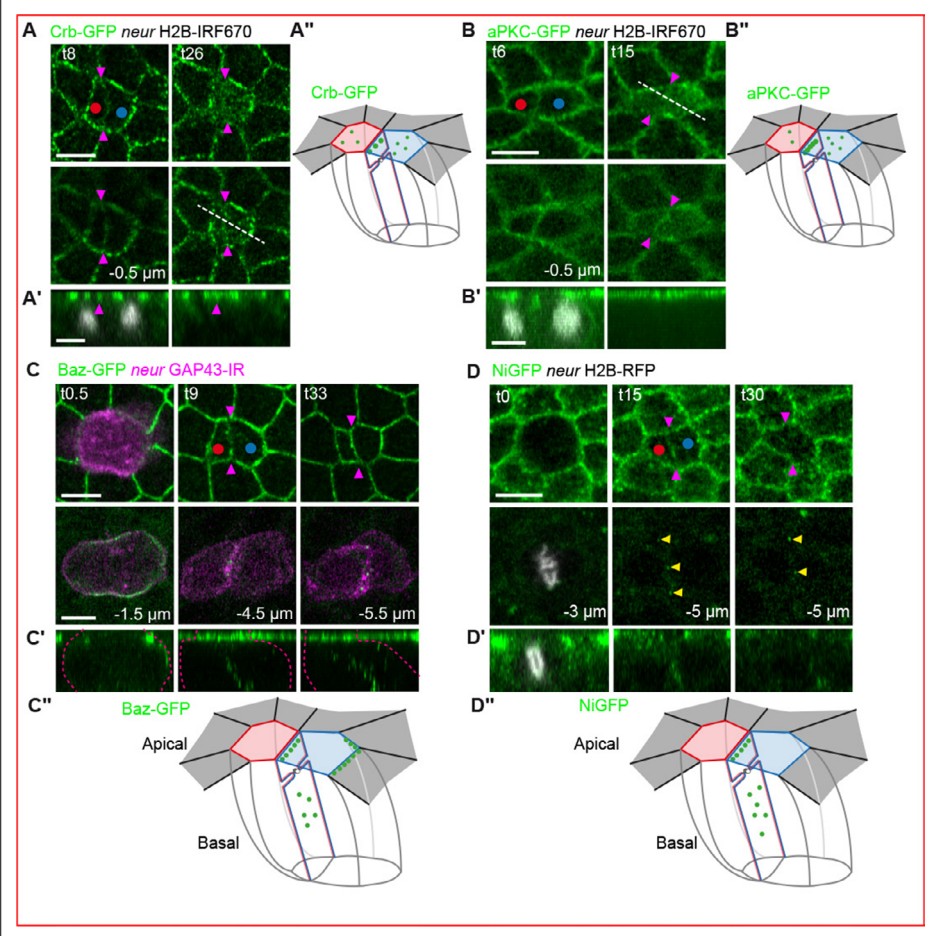

**Figure 1.** Distribution of polarity markers and Notch during sensory organ precursor (SOP) cell division. (**A–D"**) Time-lapse imaging of Crumbs (Crb)-GFP (**A, A'**, n = 22), atypical protein kinase C (aPKC)-GFP (**B, B'**, n = 10), Bazooka (Baz)-GFP (**C, C'**, n = 28), and NiGFP (**D, D'**, n = 25) during SOP cytokinesis. SOPs and their daughter cells are identified by the nuclear markers Histone 2B (H2B)-IRF670 (grey, **A and B**) and H2B-RFP (grey, **D**) or by the membrane marker growth-associated protein 43 (GAP43)-IR (magenta, **C**) expressed under the *neur* minimal driver. Top views are depicted in **A**, **B**, **C**, and **D** while the orthogonal views showing the new pIIa-pIIb interface (magenta arrowheads) are depicted in A', B', C', and D'. White dashed lines at t26 (**A**) and t15 (**B**) delineate highlight where plot profiles presented in *Figure 1—figure supplement 1C' and D'* have been performed. Red and blue dots correspond to the anterior pIIb cell and the posterior pIIa cell, respectively. Yellow arrowheads point to NiGFP lateral clusters. Magenta dashed lines delineate the SOP daughters' cell membrane. 3D schematic representations of the different proteins analysed are depicted in green in A", B", C", and D". Apical surface of the pIIb is in red while the apical surface of the pIIa is in blue. The pIIa-pIIb interface is outlined in magenta and the apical surface of neighbouring epidermal cells is outlined in dark grey. Time is in min. t0 corresponds to the onset of anaphase. Scale bars are 5 μm.

The online version of this article includes the following figure supplement(s) for figure 1:

**Figure supplement 1.** Distribution of polarity markers and Notch during sensory organ precursor (SOP) and epidermal cell divisions.

As epithelial cells are tightly packed, we first determined the origin of the Notch signal present at interface of SOP daughter cells. Indeed, during epithelial cytokinesis, the dividing cell maintains membrane contacts with the neighbouring cells, forming a *ménage à quatre* that is progressively resolved as the cell progresses towards abscission (*Daniel et al., 2018*; *Founounou et al., 2013*; *Guillot and Lecuit, 2013*; *Herszterg et al., 2013*; *Morais-de-Sa and Sunkel, 2013*; *Wang et al., 2018*). Because of its duration, the cell contact is particularly noticeable within the plane of SJs where epidermal cells maintain contact in the form of finger-like protrusions connected to the SOP midbody (t5, *Figure 1—figure supplement 1B*), until the entire belt of SJ is reformed (*Daniel et al., 2018*).

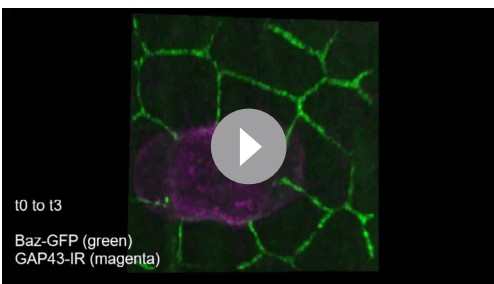

**Video 1.** 3D viewing of the time-lapse of Baz-GFP (green) and GAP43-IR (magenta) from t0 to t18, illustrating the position of the Baz-positive lateral clusters along the pIIa-pIIb interface at t18. https://elifesciences.org/articles/66659/figures#video1

To determine whether the detected NiGFP signal belongs to the pIIa-pIIb interface and not to the adjacent neighbours, we analysed the borders of clones of cells expressing NiGFP (*Figure 2A–B'*). When an SOP expressing NiGFP is dividing next to epidermal cells expressing untagged Notch (*Figure 2A*), the NiGFP signal is detected at the apical pIIa-pIIb interface, and basally between the pIIa-pIIb nuclei (*Figure 2A'* and t21). In the converse situation, when a SOP expressing untagged Notch is dividing next to epidermal cells expressing NiGFP, no GFP signal is detected at the apical and basal pIIb-pIIb interface (*Figure 2B–B'*). Analyses of clone boundaries also reveal that low NiGFP signal is detected at the boundary between pIIa or pIIb and their neighbouring epidermal cells. This is not observed in epidermal cells where Notch equally partitions along the plasma membrane (*Figure 2A' and B'*, and *Figure 1—figure supplement 1G*). These data show that, following SOP division, Notch is preferentially transported towards or stabilised at the pIIa-pIIb interface where signalling takes place.

As Notch resembles Baz localisation at the pIIa-pIIb interface, we investigated their localisation by simultaneously co-imaging NiGFP with Ubi-Baz-mCherry (*Bosveld et al., 2012*). We first observed that Ubi-Baz-mCherry colocalises with NiGFP in punctae along the pIIa-pIIb apical interface as well as in lateral interface clusters (*Figure 2C*). The NiGFP/Baz lateral interface clusters do not correspond to spot AJs (SAJs) (*McGill et al., 2009*), as E-Cad and Baz do not colocalise at the lateral pIIa-pIIb interface (*Figure 2D*). We next monitored the dynamics of Ubi-Baz-mCherry and NiGFP clusters using high spatio-temporal resolution acquisitions. Kymographs of these acquisitions (*Figure 2E and E'*) show that, at the apical pIIa-pIIb interface and even more markedly at the lateral interface, the Ubi-Baz-mCherry and NiGFP tracks colocalise to a greater extent compared with the epidermal-epidermal interface (*Figure 2E–E''*). This raises the possibility that Baz and Notch act together in space and time to contribute to pIIa/pIIb identities, which we next investigated.

## Baz contributes to Notch localisation and activation after SOP division

After having established that Ubi-Baz-mCherry/NiGFP apical and lateral interface clusters are specific to the pIIa-pIIb interface, we asked whether Notch and Baz are mutually required for cluster formation. To test if Notch is required for Baz localisation in clusters, we depleted Notch using RNAi or degradFP system (*Figure 3A–D*, *Caussinus et al., 2011*). Both approaches resulted in a reduction in Notch signal and Notch *loss-of-function* phenotypes, including an excess of SOP specification due to defective lateral inhibition, and pIIa to pIIb cell fate transformation (*Figure 3B and C*). Under these conditions, Baz still localises in clusters along the apico-basal pIIa-pIIb interface as in the wild type, indicating that Notch is dispensable for assembly of Baz clusters (*Figure 3E–F'*, yellow arrowheads).

We next test to see if, conversely, Notch localisation relies on Baz activity. Upon Baz silencing, the fluorescence intensity of the apical NiGFP clusters as well as the number and intensity of the lateral interface clusters decrease (*Figure 4A–B' and E–E'* and *Figure 4—figure supplement 2E*). A similar decrease was observed in *baz*[EH747] clones, a genetic and protein null allele of Baz (*Figure 4—figure supplement 1C,D'*; *Shahab et al., 2015*). Interestingly, the decrease of apical and lateral interface cluster number and fluorescence intensity is accompanied by a partial defect of Notch activation upon silencing of Baz (*Figure 4—figure supplement 2A, B'*), as well as in *baz*[EH747] homozygous mutant SO (*Figure 4—figure supplement 1E, E'*). Collectively, these results indicate that Baz, while not completely essential, is required for proper activation of Notch signalling in the pIIa cell.

Because Baz is deemed necessary but not sufficient per se for the assembly of Notch clusters, we hypothesised that Baz activity is required to define a threshold for Notch activation. In this model, *baz loss-of-function* would sensitise the ability of SOP daughters to signal. To test this prediction, we then asked which key regulators of Notch-dependent binary fate acquisition contribute to the assembly, dynamics, and/or signalling capacity of Baz/Notch clusters.

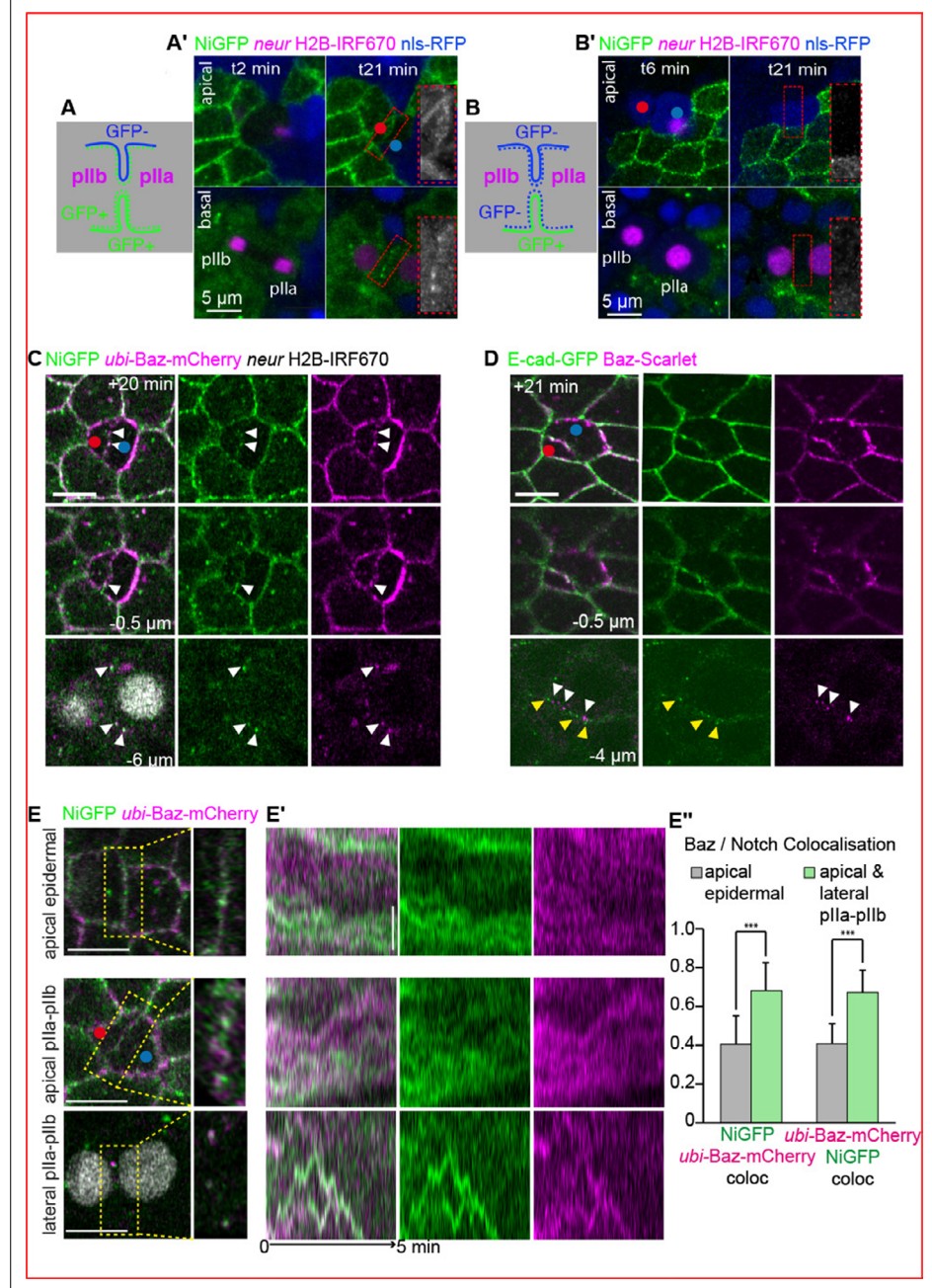

**Figure 2.** Dynamics of colocalisation of NiGFP and Bazooka (Baz)-mCherry at pIIa-pIIb interface. (**A–A'**) Schematic representation and time-lapse imaging of a dividing sensory organ precursor (SOP) expressing NiGFP (green) adjacent to epidermal cells expressing untagged version of Notch (n = 3). (**B–B'**) Schematic representation and time-lapse imaging of a dividing SOP expressing untagged version of Notch adjacent to epidermal cells expressing NiGFP (green, n = 5). Dashed and continuous lines represent the plasma membrane of SOP daughters and epidermal cells, respectively. Insets highlighted in red-dashed rectangles in A' and B' correspond to the apical and lateral interface at t21. SOPs and their daughter cells are identified by Histone 2B (H2B)-IRF670 expressed under the *neur* minimal driver (magenta). Clones of cells expressing untagged version of Notch are identified by the presence of the nuclear marker nls-RFP. Red and blue dots correspond to pIIb and pIIa daughter cells, respectively. (**C**) Localisation of NiGFP (green) together with *ubi*-Baz-mCherry (magenta) at t20 during SOP cytokinesis (n = 9). White arrowheads point to Baz- and Notch-positive clusters at the apical and lateral interface. Red and blue dots correspond to pIIb and pIIa daughter cells, respectively. (**D**) Localisation of E-cad-GFP (green) together with Baz-Scarlet (magenta) at t21 during SOP cytokinesis (n = 9). White and yellow arrowheads point to Baz-positive clusters and E-cad-positive clusters at the interface, respectively. Red and blue dots correspond

*Figure 2 continued on next page*

*Figure 2 continued*

to pIIb and pIIa daughter cells, respectively. (**E, E'**) Kymographs (**E'**) generated from high-resolution acquisitions (every 2 s, **E**) of epidermal-epidermal interface (upper panel, n = 10) or pIIa-pIIb interface (middle and lower panels) (n = 10). NiGFP is in green and *ubi*-Baz-mCherry is in magenta. Yellow-dashed rectangles highlight the position where the kymographs have been performed. Apical acquisitions have been taken around t12 and lateral acquisitions around t20. On the kymographs, tracks correspond to the movement of the clusters. SOPs and their daughter cells are identified by H2B-IRF670 expressed under the *neur* minimal driver (grey, **C and E**). In **D**, SOPs were identified based on the posterior crescent of Baz in prometaphase. (**E"**) Histogram representing the NiGFP/Baz-mCherry colocalisation (Mander's coefficient) based on kymographs (n = 10 for epidermal, n = 10 for SOP). For epidermal cells, only apical acquisitions have been considered, while apical and lateral acquisitions have been considered for pIIa-pIIb interface. ***p-value ≤ 0.001. Time is in min. t0 corresponds to the onset of anaphase. Scale bars are 5 μm in **A'**, **B'**, **C**, and **D** and 1 μm in **E'**.

## Assembly and stability of Baz-Notch clusters at pIIa-pIIb interface are modulated by Delta and Neur

We next investigated whether the activity of the Notch ligand Delta is required for the presence of Baz-Notch clusters at the pIIa-pIIb interface. A higher number of and brighter lateral interface clusters of NiGFP, accompanied with an increased transient NiGFP apical level, were observed upon silencing of Delta (*Figure 4A, A', C–C", E and E'*). However, it remains that silencing of Delta, as that of Baz alone, has a limited effect on binary cell fate acquisition, with a tufting phenotype and partial pIIa to pIIb transformations upon silencing of Delta and Baz, respectively (*Figure 4—figure supplement 2A, C'*). These low penetrant phenotypes prompted us to investigate the impact of simultaneous silencing of Baz and Delta. We found that their concomitant silencing leads to a strong neurogenic phenotype resulting a bald cuticle, that is, a penetrant loss of Notch function (*Figure 4—figure supplement 2A, A' and D, D'*). Strikingly, under this experimental situation, NiGFP was no longer stabilised in clusters at the apical or lateral interface (*Figure 4D–E'*), indicating that both Delta and Baz cooperate to stabilise Notch clusters at the pIIa-pIIb interface.

As a proxy for Delta dynamics, because Delta is hardly detected at the plasma membrane in the control situation unless its Neur-mediated endocytosis is prevented (*Trylinski et al., 2017*), we next investigated the localisations and functions of Neur. As previously reported, Neur-GFP (*Perez-Mockus et al., 2017*) localises asymmetrically at the anterior cortex, opposite to Baz during SOP prometaphase, and is unequally partitioned in pIIb cell (*Figure 5A*). At t21 min, Neur is localised in clusters at the pIIa-pIIb interface where it largely colocalises with Baz-Scarlet (*Figure 5A* and insets). Loss of Neur results in a *Notch loss-of-function* phenotype (*Lai and Rubin, 2001*) and an increased transient signal of NiGFP at the apical pIIb-pIIb-like interface (*Figure 5B–C*) accompanied with a higher number and brighter lateral interface clusters of NiGFP (*Figure 5B, B', C' and C"*), which persists for more than 36 min after anaphase onset. While upon loss of Neur, Baz still localises at the apical interface and in lateral interface clusters, the number of Baz-positive lateral cluster is significantly increased upon silencing of Neur (*Figure 5D–D"*). Analyses of their dynamics at high spatio-temporal resolution revealed that Baz and NiGFP remain closely associated at the apical and lateral interface clusters upon loss of Neur (*Figure 5E and E'*). Collectively, these results argue that Neur, Delta, Baz, and Notch localise in apical and lateral interface clusters and that their numbers and signal intensities depend on Neur and Delta activity, on the one hand, and on Baz activity, on the other hand. These data further suggest that clusters are assembled but fail to be disassembled in a timely manner upon Neur silencing.

## Numb negatively regulates the Baz-Notch lateral interface clusters

To further investigate the minimal requirements for the occurrence of Baz-Notch clusters at the pIIa-pIIb interface, we next analysed the function of Numb. In contrast to Neur, Numb does not colocalise with Baz-Scarlet-positive clusters at the pIIa-pIIb interface (*Figure 6A*). Inhibition of Numb, which results in a *gain-of-function* Notch phenotype (*Guo et al., 1996*), is accompanied by increased Notch transient signals at the apical pIIa-pIIa-like interface (*Figure 6B–C*), and a higher number of brighter clusters at the lateral interface of NiGFP (*Figure 6B, B', C' and C"*). These data are consistent with those previously published (*Couturier et al., 2012*; *Trylinski et al., 2017*), and further show that the accumulation of lateral Notch clusters persists until at least 36 min after anaphase onset. Upon

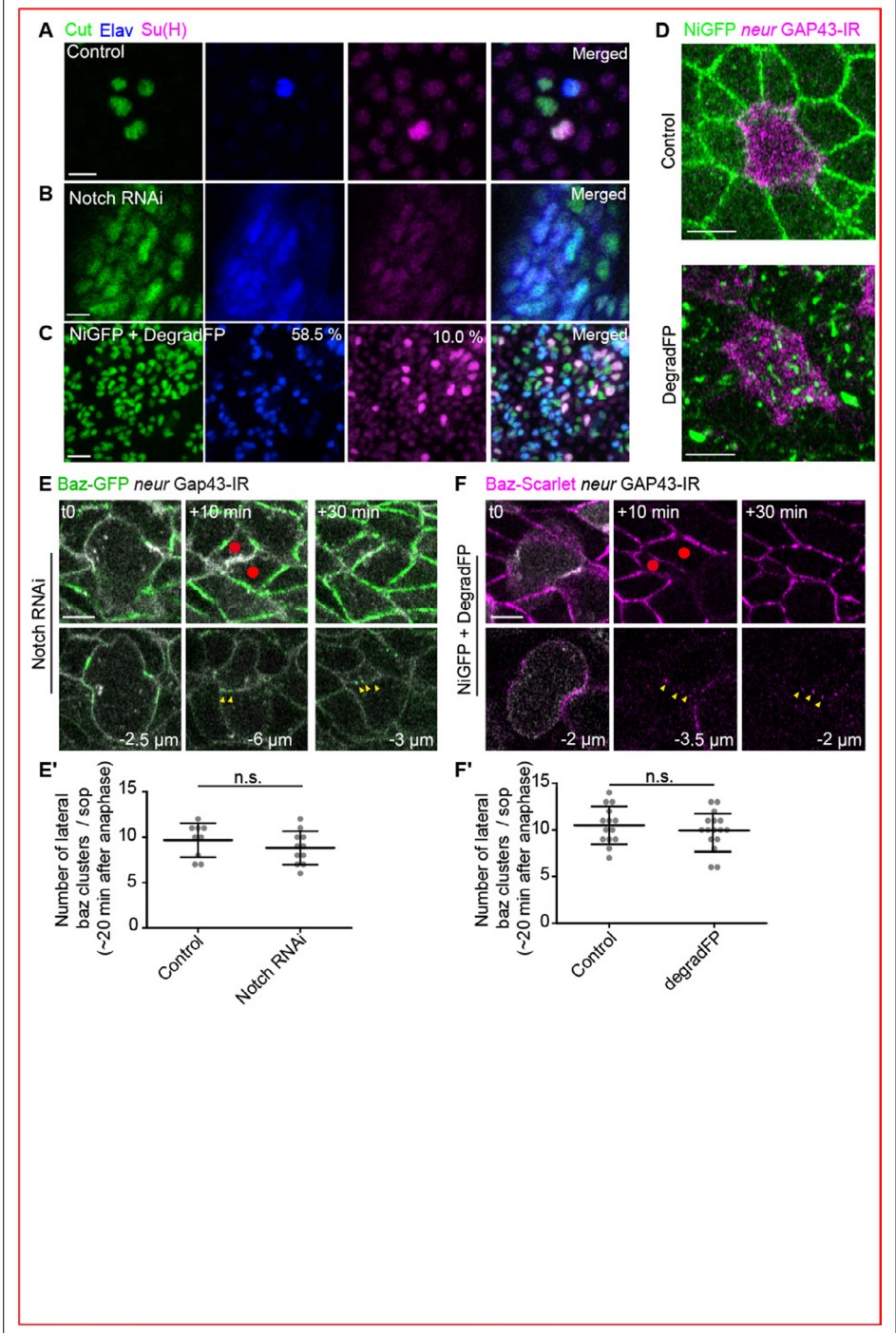

**Figure 3.** Notch *loss-of-function* does not impair the formation of Bazooka (Baz) clusters along the pIIb-pIIb-like interface. (**A–C**) SO lineage analysis using the SO marker Cut (green), the socket marker Su(H) (magenta) and the neuronal marker Elav (blue) in control lineage (**A**, n = 3 nota), upon silencing of Notch (**B**, n = 2 nota) or upon degradation of NiGFP with degradFP (**C**, n = 4 nota). Upon degradation of NiGFP by degradFP (**C**), 58.5% and 10% of Cut-positive cells are Elav or Su(H) positive, respectively (n = 1045 Cut-positive cells). In a control situation, only 25% of the Cut-positive cells are Elav or Su(H) positive. (**D**) Localisation of NiGFP (green) together with growth-associated protein 43 (GAP43)-IR (magenta) expressed under the *neur* minimal driver in control or upon NiGFP degradation by degradFP (n = 19). (**E–F**) Localisation of Baz-GFP (green in **E**) and Baz-mScarlet (magenta in **F**) together with GAP43-IR expressed under the *neur* minimal driver (grey). Yellow arrowheads point the clusters at the lateral interface. Red dots label pIIb ant pIIb-like cells. (**E'–F'**) Quantification of the number of Baz-positive

*Figure 3 continued on next page*

*Figure 3 continued*

lateral clusters at t20/21 in control (**E'**, n = 10; **F'**, n = 14) or upon Notch silencing (**E'**, n = 11) or upon NiGFP degradation by degradFP (**F'**, n = 16). ns, not statistically significant, p-value ≥ 0.05. Time is in min. t0 corresponds to the onset of anaphase. Scale bars are 5 μm.

silencing of Numb, Baz localises in lateral interface clusters at the pIIa-pIIa like interface (*Figure 6D, D' and E*), where it colocalises with NiGFP, as revealed by the dynamics of the Baz-Notch clusters at high spatio-temporal resolution (*Figure 6F*).

Together, these data indicate that Numb decreases the number of Notch-Baz clusters. As Numb is present and regulates Notch endosomal trafficking in the anterior pIIb cell (*Cotton et al., 2013*; *Couturier et al., 2013*), our data suggest that Notch-Baz clusters are assembled in the anterior cell upon loss of Numb and contribute to Notch activation in this cell. This model further suggests that

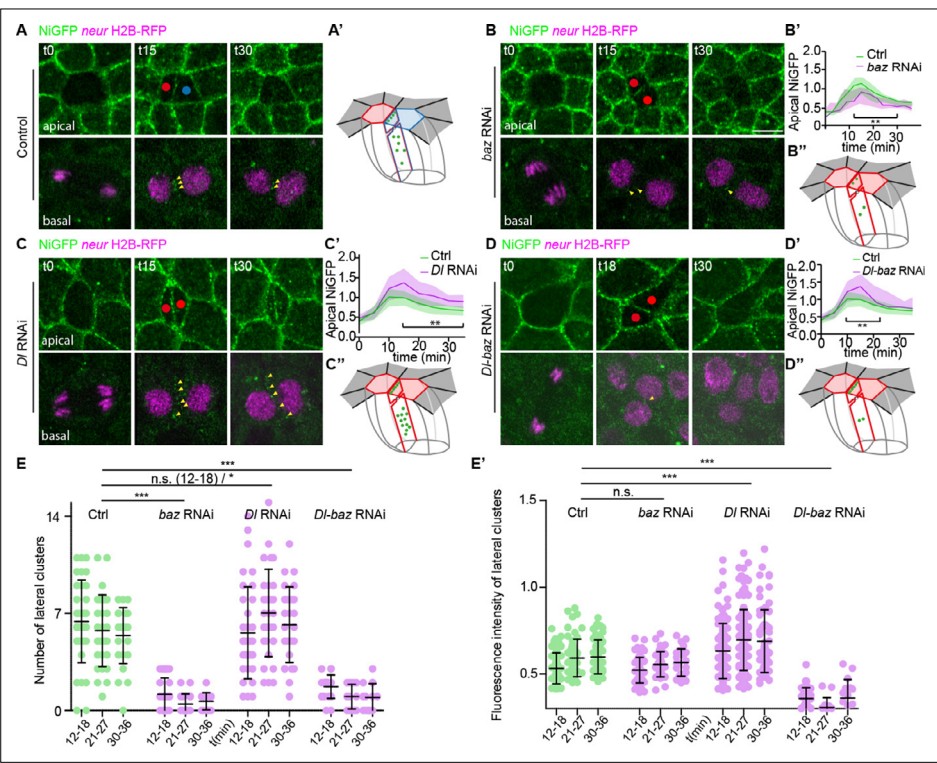

**Figure 4.** Bazooka (Baz) and Delta are required for localisation of Notch in lateral clusters and for Notch activation. (**A–D"**) Time-lapse imaging of NiGFP (green) together with H2B-RFP expressed under the *neur* minimal driver (magenta) during sensory organ precursor (SOP) cytokinesis in control (**A**), upon silencing of Baz (**B**), Delta (**C**), or both (**D**). Yellow arrowheads point to NiGFP-positive lateral clusters. Red and blue dots correspond to the pIIb or pIIb-like cells and pIIa cell, respectively. 3D schematic representations of NiGFP localisation (green) along the interface in the different genetic contexts are depicted in **A', B", C", and D"**. Apical surface of pIIb and pIIb-like cells is in red while the apical surface of the pIIa cellis in blue. The interface between the SOP daughters is outlined in magenta (**A'**) or in red (**B", C", and D"**) and the apical surface of neighbouring epidermal cells is outlined in dark grey. Quantification of NiGFP fluorescence intensity at the apical interface between SOP daughters are shown in **B', C', and D'**. Control (n = 11) is in green while *baz* RNAi (n = 11), *Delta* RNAi (n = 13), and *baz, Delta* RNAi (n = 14) are in magenta. ns, not statistically significant, p-value > 0.05 and **p-value ≤ 0.01. (**E–E'**) Quantification of the number (**E**) and fluorescence intensity (**E'**) of NiGFP-positive lateral clusters over time in control (green, n = 11 or upon silencing of Baz [n = 11], Delta [n = 13], or both [n = 14] RNAi in magenta). ns, not statistically significant, p-value ≥ 0.05, *p-value < 0.05 and ***p-value ≤ 0.001. Time is in min. t0 corresponds to the onset of anaphase. Scale bars are 5 μm.

The online version of this article includes the following figure supplement(s) for figure 4:

**Figure supplement 1.** NiGFP localisation in *baz*[EH747] mutant clones.

**Figure supplement 2.** Requirements for Delta and Bazooka (Baz) for Notch activation.

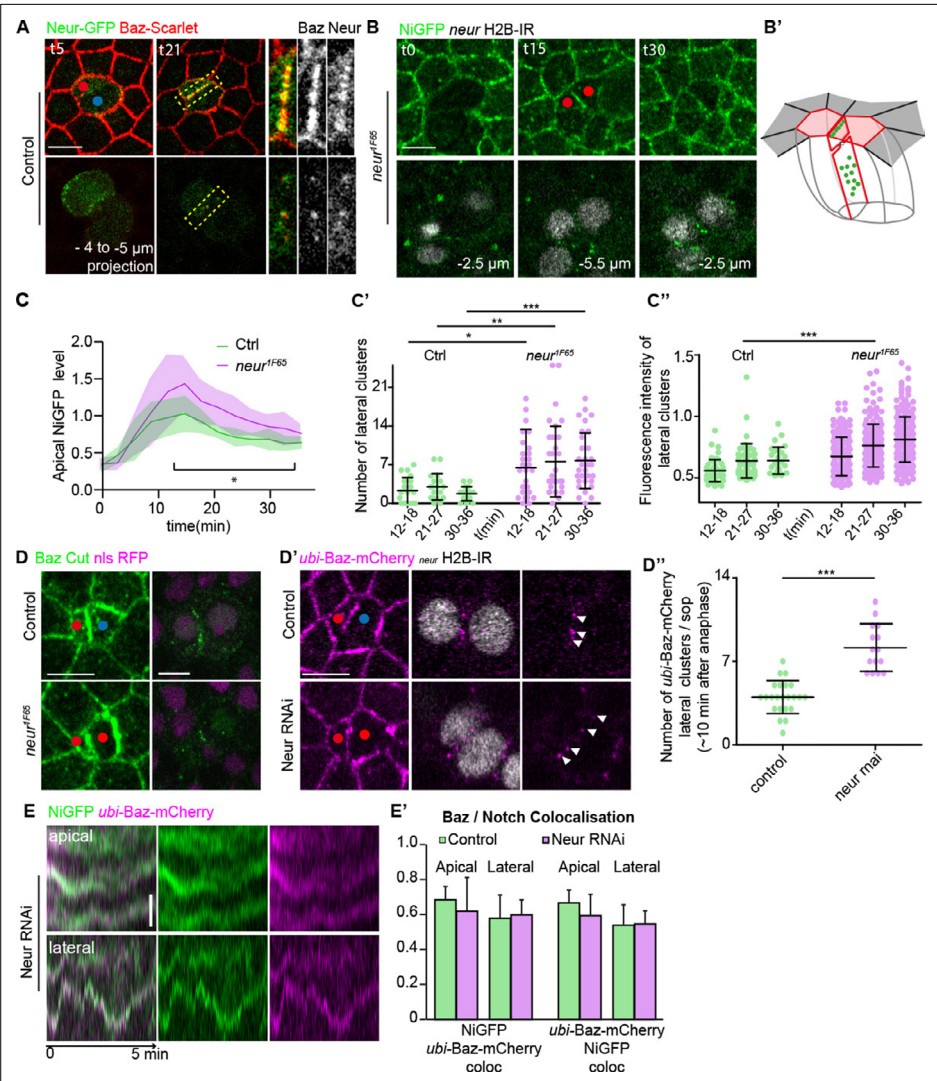

**Figure 5.** Neuralized (Neur) localises in and regulates the number of NiGFP/Bazooka (Baz)-positive clusters. (**A**) Time-lapse imaging of Neur-GFP (green) together with Baz-Scarlet (red) during sensory organ precursor (SOP) cytokinesis in control (n = 8). Yellow dashed rectangles highlight the high magnifications depicted on the panels on the right. (**B**) Time-lapse imaging of NiGFP (green) together with Histone 2B (H2B)-RFP expressed under the *neur* minimal driver (grey) during SOP cytokinesis in *neur$^{1F65}$* (n = 13). Red and blue dots correspond to the pIIb or pIIb-like cells and pIIa, respectively. Clones of mutant cells are identified by the loss of the nuclear marker nls-RFP. (**B′**) 3D schematic representations of NiGFP localisation (green) along the interface in *neur$^{1F65}$*. Apical surface of pIIb and pIIb-like cells is in red. The interface between the SOP daughters is outlined in red and the apical surface of neighbouring epidermal cells is outlined in dark grey. (**C**) Quantification of NiGFP fluorescence intensity at the apical interface between SOP daughters in control (green, n = 9) and in *neur$^{1F65}$* mutant (magenta, n = 10). *p-value ≤ 0.05. (**C–C″**) Quantification of the number (**C′**) and fluorescence intensity (**C″**) of NiGFP-positive lateral clusters over time in control (green, n = 9) or in *neur$^{1F65}$* mutant (magenta, n = 10). *p-value < 0.05, **p-value ≤ 0.01, and ***p-value ≤ 0.001. (**D**) Localisation of Baz (green) in control (n = 7) *neur$^{1F65}$* mutant (n = 7). Clones of mutant cells are identified by the loss of the nuclear marker nls-RFP (magenta). (**D′**) Localisation of Baz-mCherry (magenta) together with H2B-IRF670 (grey) expressed under the *neur* minimal driver in control (n = 23) and upon silencing of Neur (n = 15) during SOP cytokinesis. Pictures show the SOP daughter cells 10 min after the onset of anaphase. White arrowheads point the Baz-positive clusters at the lateral interface. Red and blue dots correspond to the pIIb or pIIb-like cells and pIIa, respectively. (**D″**) Quantification of the number of Baz-positive clusters at the lateral interface in control (green, n = 23) or upon silencing of Neur (magenta, n = 15). ***p-value ≤ 0.001. (**E**) Kymographs generated from high-resolutions acquisition (every 2 s, starting at t12 min) and illustrating the colocalisation between NiGFP (green) and Baz-mCherry (magenta) at the interface of SOP daughters upon silencing of Neur (n = 11). On the kymographs, tracks correspond to the movement of the clusters. (**E′**) Histogram representing the

*Figure 5 continued on next page*

*Figure 5 continued*

NiGFP/Baz-mCherry colocalisation (Mander's coefficient) in apical and lateral clusters based on kymographs in control (green, n = 10) and upon silencing of Neur (magenta, n = 6). Time is in min. t0 corresponds to the onset of anaphase. Scale bars are 5 and 1 μm for the kymographs.

---

Numb acts antagonistically to Baz to promote Notch clusters assembly and/or stability. To test this prediction, we overexpressed Numb in the SOP and daughter cells and observed that NiGFP is no longer detected along the pIIb-pIIb-like interface, either apically or laterally (*Figure 6D''*). While Baz localises uniformly at the apical SOP daughter cell interface, lateral interface clusters are barely detectable (t14, *Figure 6D''*, bottom panels, and E). These data raise the possibility that Numb and Baz act antagonistically, as has been proposed in vertebrates (*Nishimura and Kaibuchi, 2007*; *Sun et al., 2016*). As Numb interacts with the NPAF motif of Spdo to control Notch/Spdo endosomal trafficking, the above data call into question the relationship between Baz and Spdo, which we studied next.

## Spdo is required for Baz-Notch cluster formation

On live specimens, Baz-Scarlet and Spdo-GFP (*Couturier et al., 2013*) colocalise both at the apical pIIa-pIIb interface and in lateral clusters (*Figure 7A*, t21). Compared with the control situation, Baz-positive lateral clusters are no longer detectable upon loss of Spdo (*Figure 7B* and *Figure 7—figure supplement 1C*). In agreement with the findings of *Couturier et al., 2012*, loss of Spdo also results in an increase of NiGFP signal at the apical interface of SOP daughters and the appearance of a continuous and nebulous staining of NiGFP instead of the characteristic, well-defined, lateral clusters observed at the pIIa-pIIb interface of controls SO (*Figure 7C and D* and *Figure 7—figure supplement 1A-A',C-C'*). We also noticed that NiGFP persists at the apical interface compared with the control, and that NiGFP is detected apically, in the cytoplasm or at the apical plasma membrane, indicative of higher levels of Notch upon loss of Spdo (*Figure 7C,D*,t15 and t30, upper panels). Fluorescence recovery after photobleaching (FRAP) analyses revealed that the NiGFP signal at the apical interface is recovered 1.9 times faster, with a mobile fraction 1.6 times higher than in the control situation (*Figure 7E and E'* and *Figure 7—figure supplement 1B*). The changes in NiGFP distribution and time residence at the pIIb-pIIb-like interface are accompanied by a loss of colocalisation of NiGFP and Baz-mCherry at the apical and lateral pIIb-pIIb-like interface upon Spdo silencing (*Figure 7—figure supplement 1C,D'*). We first concluded that Spdo co-clusters with Baz and Notch at the pIIa-pIIb interface and, second, that the activity of Spdo is required for the clustering of Baz/Notch along the pIIa-pIIb interface to promote Notch activation.

## Discussion

In this study, we have characterised the remodelling of apico-basal cell polarity occurring during SOP division leading to a specific pIIa-pIIb Notch signalling interface. We report that Baz, but not aPKC, co-partitions with Notch, Spdo, and Neur in apical and lateral clusters. The assembly of these clusters requires Baz and Spdo activities, and their number and dynamics are regulated by Delta, Neur and Numb activities. In the absence of Numb, the number of clusters increases, while overexpression of Numb results in their disappearance, suggesting that Numb and Baz act antagonistically. We propose a model according to which Notch/Baz/Spdo/Neur clusters represent the Notch signalling units at the pIIa-pIIb interface.

## The pIIa-pIIb interface possesses an atypical apico-basal polarity compared with epidermal cells

Previous pioneer work has determined that in dividing SOPs, Par3/Par6/aPKC and Pins/Dlg polarity modules are specifically relocated from the apico-basal cortex into the posterior-anterior cortex, respectively (*Bellaïche et al., 2001*; *Roegiers et al., 2001*). Assembly of the Baz/Par6/aPKC complex is initiated by the phosphorylation of Par6 by the mitotic kinase AurA (*Wirtz-Peitz et al., 2008*). Here, we report that during cytokinesis coinciding with the presumptive proteolytic degradation of AurA, the Baz /Par6/aPKC complex disassembles with aPKC redistributing like Crb in apical intracellular compartments. This is at the expense of its regular plasma membrane localisation observed in epidermal cells. In contrast to aPKC, Baz redistributes apically at the posterior pole of the pIIa cell and

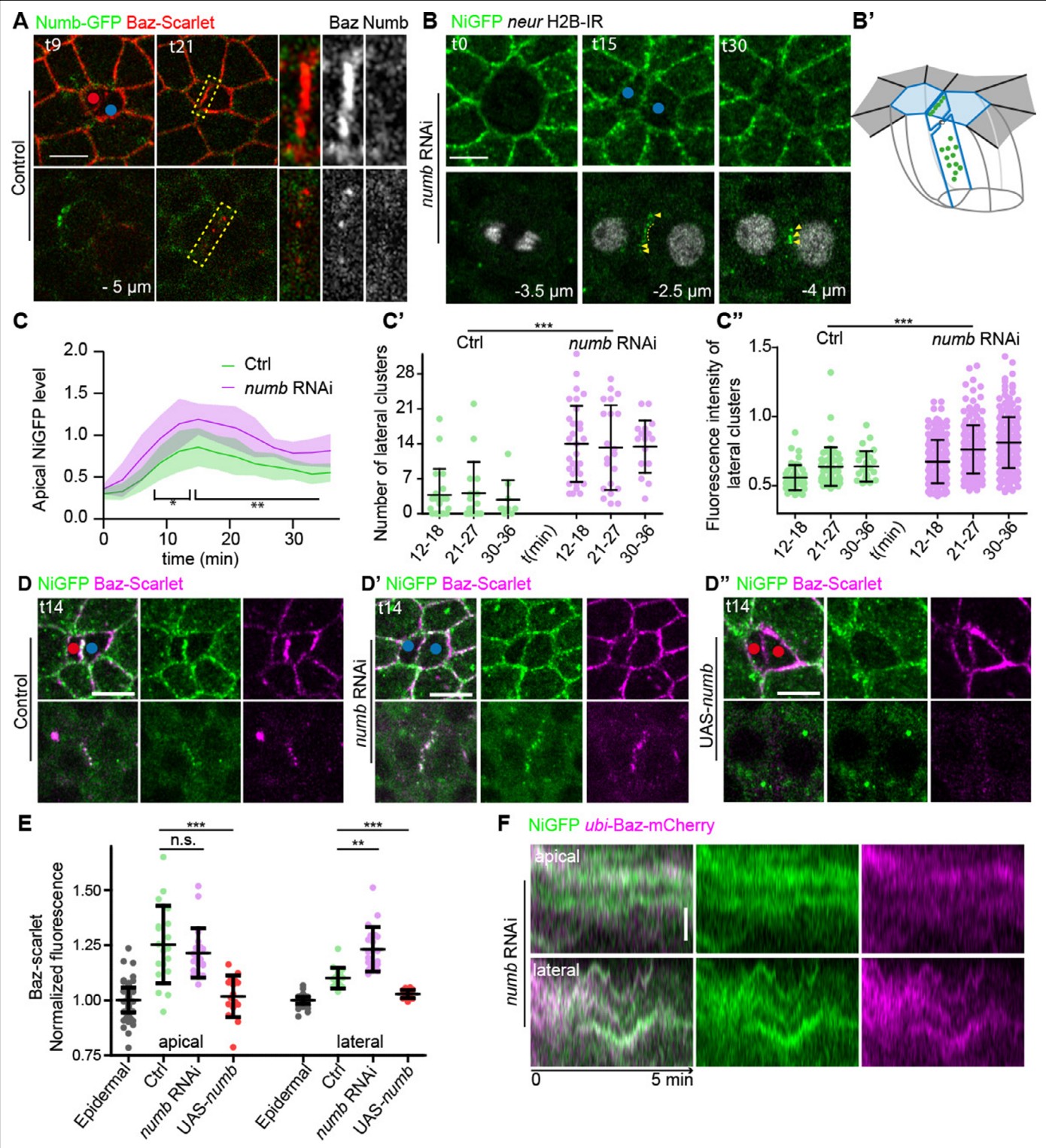

**Figure 6.** Numb regulates the number of NiGFP/Bazooka (Baz)-positive clusters. (**A**) Time-lapse imaging of Numb-GFP (green) together with Baz-Scarlet (red) during sensory organ precursor (SOP) cytokinesis in control (n = 5). Yellow-dashed rectangles highlight the high magnifications depicted on the panels on the right. (**B**) Time-lapse imaging of NiGFP (green) together with Histone 2B (H2B)-IRF670 expressed under the *neur* minimal driver (grey) during SOP cytokinesis upon the silencing of Numb (n = 20). Yellow arrowheads and yellow dashed line point to the lateral clusters. Red and blue dots correspond to the pIIb cells and pIIa or pIIa-like cells, respectively. (**B'**) 3D schematic representations of NiGFP localisation (green) along the interface upon silencing of Numb. Apical surface of pIIa and pIIa-like cells is in blue. The interface between the SOP daughters is outlined in blue and the apical

*Figure 6 continued on next page*

*Figure 6 continued*

surface of neighbouring epidermal cells is outlined in dark grey. (**C**) Quantification of NiGFP fluorescence intensity at the apical interface between SOP daughters in control (green, n = 11) and upon silencing of Numb (magenta, n = 20). *p-value < 0.05 and **p-value ≤ 0.01. (**C'–C"**) Quantification of the number (**C'**) and fluorescence intensity (**C"**) of NiGFP-positive lateral clusters over time in control (green, n = 11) or upon silencing of Numb (magenta, n = 20). ***p-value ≤ 0.001. (**D–D"**) Localisation of NiGFP (green) together with Baz-Scarlet (magenta) at t14 during SOP cytokinesis in control (**D**, n = 14), upon silencing of Numb (D', n = 9) or upon overexpression of Numb (D", n = 8). Red and blue dots correspond to the pIIb-like cells and pIIa or pIIa-like cells, respectively. (**E**) Quantification of the fluorescence intensity of Baz-Scarlet at the apical or lateral interface of control epidermal cells (grey, n = 31), of SOP daughters in the control (green, n = 14), upon silencing of Numb (magenta, n = 9) or upon overexpression of Numb (red, n = 8). ns,not statistically significant, p-value ≥ 0.05, **p-value ≤ 0.01, and ***p-value ≤ 0.001. (**F**) Kymographs generated from high-resolution acquisition (every 2 s) and illustrating the colocalisation between NiGFP (green) and Baz-mCherry (magenta) at the interface of SOP daughters upon silencing of Numb (n = 14). On the kymographs, tracks correspond to the movement of the clusters. Time is in min. t0 corresponds to the onset of anaphase. Scale bars are 5 and 1 μm for the kymographs.

in the form of clusters at the apical and lateral pIIa-pIIb interface. Such lateral clusters of Baz are only found at the pIIa-pIIb interface, indicating that the SOP-specific remodelling polarity that takes place at SOP mitosis is instrumental in formation of clusters. Baz has been reported to be excluded from the lateral plasma membrane following Par1-mediated phosphorylation (*Benton and St Johnston, 2003*). In addition, phosphorylation of Baz by Par1 activity is antagonised by type 2 A protein phosphatase (PP2A) activity (*Krahn et al., 2009*), and silencing of *tws*, the regulatory B subunit of PP2A, results in *Notch gain-of-function* phenotype (*Shiomi et al., 1994*). It is as yet unclear how SOP polarity remodelling leads to Baz cluster assembly and lateral localisation. The fact that Spdo and Notch, two transmembrane proteins, colocalise with Baz in lateral clusters (both on fixed and live specimens) argues against a model according to which N-terminal oligomerisation domain of Baz could drive phase separation of Baz (*Liu et al., 2020*) at this location.

The nanoscopic clusters of Baz are reminiscent of the clusters serving as an AJ assembly landmark, by repositioning Cadherin-Catenin clusters at apico-lateral sites for assembly of SAJ (*McGill et al., 2009*). The Baz and Cadherin-Catenin clusters are shown to assemble independently and the number and size of Cadherin-Catenin clusters are decreased in the *baz* mutant as reported here for Notch clusters. This indicates a common function of Baz in controlling the assembly, positioning, and stability of clusters. By analogy, it can be proposed that Notch/Spdo trans-interacting with Delta forms clusters independently of Baz clusters assembled through Baz oligomerisation, and that Baz is required to position the cluster at the correct localisation along the pIIa-pIIb interface.

In addition to organising membrane nanoscopic clusters of Cadherin and Catenin, in vertebrates, Par3 also functions as a receptor for exocyst, a protein complex of the secretory pathway required for the delivery of basolateral proteins to the plasma membrane (*Ahmed and Macara, 2017*). It is interesting to note that Baz clusters are exclusively located at the pIIa-pIIb interface. Analyses of NiGFP clone borders have revealed a preferential localisation of Notch at the pIIa-pIIb interface instead of being equally partitioned at the plasma membrane. Together with the fact that Sec15, a component of the exocyst complex, regulates Notch and Spdo trafficking to regulate binary fate acquisition in the SO lineage (*Jafar-Nejad et al., 2005*), our results place Baz as a potential regulator in the delivery of Notch/Spdo at specific sites along the pIIa-pIIb interface.

In any case, while Baz activity is required for efficient Notch cluster assembly, only a limited number of cell fate transformations are observed upon loss of Baz. We propose a model whereby Baz activity is required to define a threshold for Notch activation and Baz *loss-of-function* sensitises the ability of SOP daughters to signal. In favour of this model, the concomitant silencing of Delta and Baz, which individually induce a low rate of cell fate transformations, leads to a complete Notch *loss-of-function* phenotype. Whether Baz and Delta act in the same signalling units (see below) or not remains to be determined. An alternative possibility is that Baz functions together with Serrate, the second Notch ligand. Indeed, silencing of Delta leads to a strong lateral inhibition defect, but the function of Dl during asymmetric cell division can be substituted almost completely by *Ser*, explaining the tufting phenotype upon Delta silencing (*Zeng et al., 1998*). If Baz regulates Serrate activity, then simultaneous silencing of Baz and Delta would lead to a complete loss of ligand activity, hence Notch *loss-of-function*. It is worth noting that in the *Drosophila* optic lobe, Notch, Serrate, and Canoe have been shown to form a complex (*Perez-Gomez et al., 2013*), and Baz, by virtue of it regulating Canoe localisation (*Choi et al., 2013*), may regulate the distribution and/or activity of such a complex. Whatever

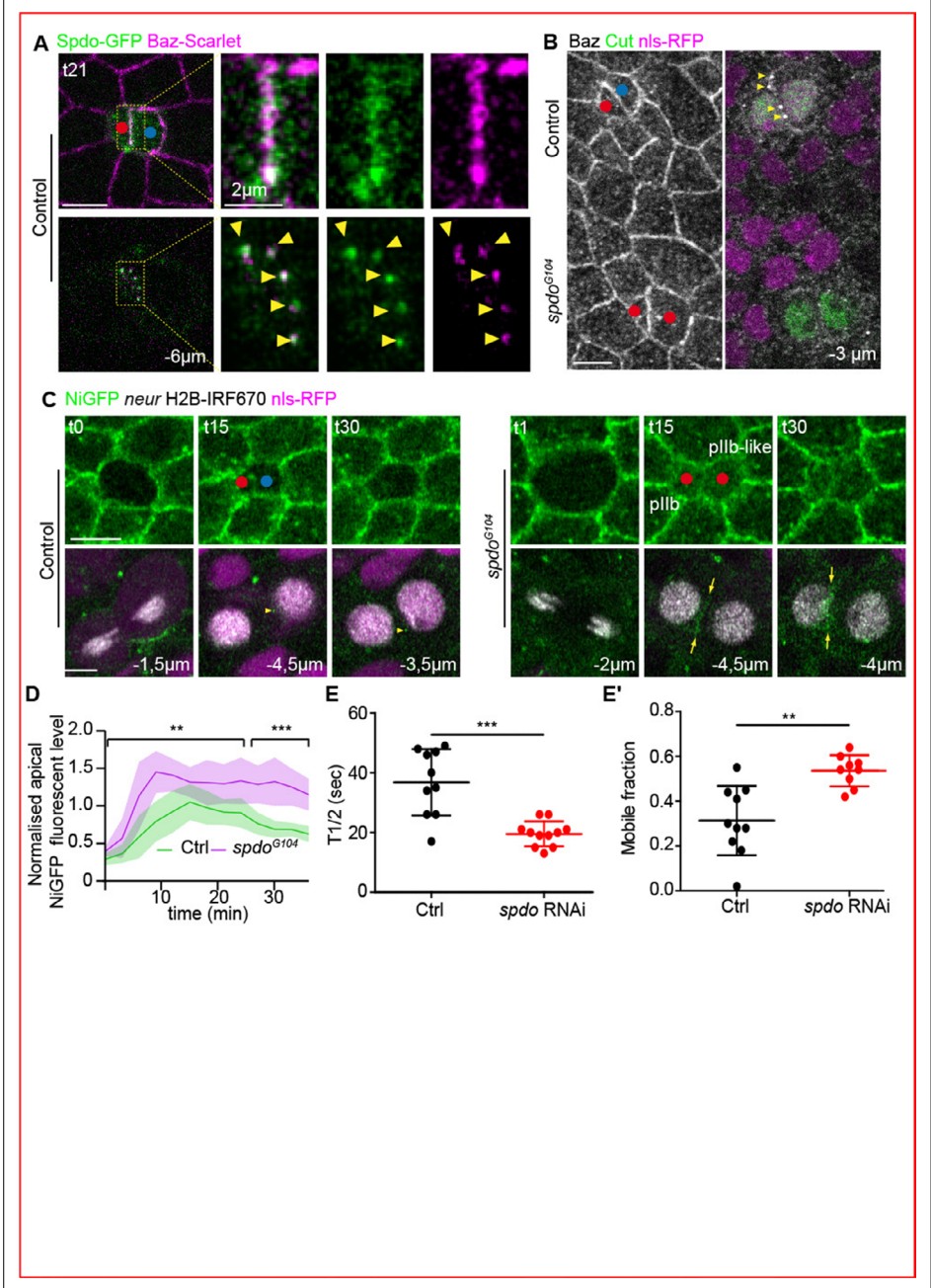

**Figure 7.** Sanpodo is required for the assembly of NiGFP/Bazooka (Baz) clusters. (**A**) Localisation of Sanpodo (Spdo)-GFP (green) together with Baz-Scarlet (magenta) at t21 during sensory organ precursor (SOP) cytokinesis in control (n = 10). Yellow-dashed rectangles highlight the high magnifications depicted on the panels on the right. Yellow arrowheads point to lateral clusters positive for Baz and Spdo. (**B**) Localisation of Baz (anti-N term, grey) together with the SO marker Cut (green) in control (n = 20 SOP ) or *spdo^G104* mutant clones (n = 25 SOP). Yellow arrowheads point to Baz-positive clusters at the lateral interface. (**C**) Time-lapse imaging of NiGFP (green) together with Histone 2B (H2B)-IR (grey) expressed under the *neur* minimal driver (grey) during SOP cytokinesis in control (n = 14) or in *spdo^G104* mutant (n = 10). Yellow arrowheads point to NiGFP-positive clusters at the lateral interface while the yellow arrows point to the NiGFP continuous signal along the lateral interface of SOP daughters. *spdo^G104* mutant clones are identified by the loss of the nuclear marker nls RFP (magenta). Red and blue dots correspond to the pIIb or pIIb-like cells and pIIa cells, respectively. (**D**) Quantification of NiGFP fluorescence intensity at the apical interface between SOP daughters in control (green, n = 14) and *spdo^G104* mutant (magenta, n = 10). **p-value ≤ 0.01 and ***p-value ≤ 0.001. (**E–E'**) Quantification of the t1/2 (**E**) and the mobile fraction (**E'**) of NiGFP following fluorescence recovery after photobleaching (FRAP) at the apical interface of SOP daughters in control (black, n

*Figure 7 continued on next page*

*Figure 7 continued*

= 10) or upon silencing of Spdo (red, n = 11) at t20. **p-value ≤ 0.01 and ***p-value ≤ 0.001. Time is in min. t0 corresponds to the onset of anaphase. Scale bars are 5 μm.

The online version of this article includes the following figure supplement(s) for figure 7:

**Figure supplement 1.** Sanpodo is required for the assembly of NiGFP/Bazooka (Baz) clusters.

the mechanism of action, Baz by regulating the size and number of clusters at the pIIa-pIIb interface appears to be important for proper Notch signalling during SOP cytokinesis.

## Do the Notch/Baz/Spdo clusters constitute signalling units?

The clusters present at the pIIa-pIIb interface are positive for Notch, Spdo, Baz, and Neur. While Delta is also detected along the pIIa-pIIb interface on fixed specimens (*Bellec et al., 2021*), DlGFP was reported to be barely detectable in living pupae unless Neur-mediated Delta endocytosis was blocked (*Trylinski et al., 2017*). This led to the proposal that newly synthesised Delta reaches the plasma membrane and signals from there thus exhibiting a rapid turnover/endocytosis. An implication of these findings is that the clusters are present on both sides of the pIIa-pIIb interface as a kind of snap button with Delta/Neur in the pIIb cell interacting in trans with Notch/Spdo in the pIIa cell. Based on the role of Numb in Notch/Spdo trafficking in the pIIb cell, the fact that Baz is enriched in the posterior pIIa cell at cytokinesis and the proposed antagonism between Numb and Baz, we anticipate that Baz is located primarily in clusters on the pIIa cell side. As the time residence of Delta, Notch, and Baz in the cluster is very short (on the time scale of minutes), it implies that Delta can interact with Notch in trans, and be internalised in a Neur-dependent manner to promote the S2 cleavage of Notch in the minute time scale. We propose that Baz-mediated clustering might be a means to concentrate Notch/Spdo locally and increase its ability to interact with Delta.

## Site of NICD production

Our study brings further support to the notion of a tight coupling between cell polarity and Notch signalling. Photobleaching and phototracking experiments during SOP cytokinesis reveal that among the two pools of Notch, the basolateral pool located basally to the midbody is the main contributor (*Trylinski et al., 2017*). While the apical pool of Notch also contributes to NICD production, it is as yet unclear whether NICD is directly produced from the apical pIIa-pIIb interface or if a basolateral relocation is a prerequisite (*Bellec et al., 2021*; *Couturier et al., 2012*; *Trylinski et al., 2017*). According to that model, NICD production would primarily occur at the lateral pIIa-pIIb interface. Our results, which show that the composition of the presumptive signalling clusters are similar at the apical and basolateral pIIa-pIIb interface, may indicate that NICD could be directly produced from both sites. The remodelling of cell polarity taking place during SOP cytokinesis could thus enable the formation of equally potent signalling clusters along the pIIa-pIIb interface, favouring private pIIa-pIIb cell-cell communication. The amounts and half-life of such signalling clusters could account for the respective contributions of basal versus apical pools in producing NICD.

## Numb and Baz act oppositely on Notch/Spdo cluster assembly

While the loss of Neur and loss of Numb both result to an increase in the number and intensity of Baz/Notch/Spdo clusters, the causes are different. Upon a lack of Neur, we anticipate that Delta is bound to Notch in trans. In the absence of Neur-mediated endocytosis of Delta that exerts pulling forces on Notch, the clusters are stabilised/not consumed. Numb interacts physically with Spdo to control the subcellular localisation of the Notch/Spdo complex. In the control situation, Numb is not detected in the Notch/Spdo clusters at the pIIa-pIIb interface, suggesting that Notch and Spdo clusters at the interface are predominantly on the pIIa side. Loss of Numb that leads to recycling of Notch/Spdo towards the plasma membrane of the pIIb cell results in an increase in the number and intensity of Notch/Spdo/Baz clusters at the pIIa-pIIb interface. By contrast, overexpression of Numb causes the disappearance of Notch/Spdo clusters at the pIIa-pIIb interface.

By analogy with vertebrates, we anticipate that Numb, due to its ability to bind to Baz (*Nishimura and Kaibuchi, 2007*), is somehow competing with Baz for access to Notch/Sdpo, and therefore formation of Notch/Spdo/Baz signalling clusters. Based on the fact that loss of Spdo leads to a stronger reduction in Baz/Notch cluster assembly, one prediction to be tested is that Baz interacts with Spdo/Notch.

## Concluding remarks

Due to the conservation of intra-lineage communication, it would be interesting to investigate whether a cell-cell communication interface exhibits an atypical apico-basal polarity and if Par3-dependent clustering of Notch also regulates private communication between daughters in vertebrates.

# Materials and methods

**Key resources table**

| Reagent type (species) or resource | Designation | Source or reference | Identifiers | Additional information |
|---|---|---|---|---|
| Gene (*Drosophila melanogaster*) | w[1118] | Bloomington *Drosophila* Stock Center | BDSC: 3,605 FLYB: FBal0018186; RRID:BDSC_3605 | |
| Gene (*Drosophila melanogaster*) | Nrg::YFP P{PTT-GA}NrgG00305 | Kyoto Stock Center *Morin et al., 2001* | FLYB: FBgn0264975;RRID:KSC_110658 | |
| Gene (*Drosophila melanogaster*) | *neur*-H2B::IR | This study | | Two lines generated (II[d] and III[d]) |
| Gene (*Drosophila melanogaster*) | Crb::GFP (A) | *Huang et al., 2009* | FLYB: FBgn0259685 | Gift from Dr Y Hong |
| Gene (*Drosophila melanogaster*) | aPKC::GFP | *Besson et al., 2015* | FLYB: FBgn0261854 | Kind gift from Dr F Schweisguth |
| Gene (*Drosophila melanogaster*) | Baz::GFP P{PTT-GC}baz[CC01941] | Bloomington *Drosophila* Stock Center *Buszczak et al., 2007* | FLYB:FBgn0000163; RRID:BDSC_51572 | |
| Gene (*Drosophila melanogaster*) | Baz::mScarlet | kind gift from Dr J Januschke | FLYB:FBgn0000163 | |
| Gene (*Drosophila melanogaster*) | *neur*-GAP43::IR | This study | | Two lines generated (II[d] and III[d]) |
| Gene (*Drosophila melanogaster*) | Dlg::GFP | Bloomington *Drosophila* Stock Center *Buszczak et al., 2007* | FLYB: FBgn0001624; RRID: BDSC_ 50859 | |
| Gene (*Drosophila melanogaster*) | *Ubi-p63E-Baz-mCherry (II) and (III)* | *Bosveld et al., 2012* | Transgenic lines; FLYB:FBgn0000163 | Kind gift from Dr Y Bellaiche |
| Gene (*Drosophila melanogaster*) | *NiGFP* | *Bellec et al., 2018* | FLYB: FBgn0004647 | CRISPR line |
| Gene (*Drosophila melanogaster*) | *neur*-H2B-RFP | *Gomes et al., 2009* | | Kind gift from Dr F Schweisguth |
| Gene (*Drosophila melanogaster*) | *baz[EH747], FRT19A/Y* | Kind gift of A Wodarz | FLYB:FBgn0000163 | |
| Gene (*Drosophila melanogaster*) | *Ubi-mRFP.nls, w*, hsFLP, FRT19A* | Bloomington *Drosophila* Stock Center | RRID:BDSC_31418 | |
| Gene (*Drosophila melanogaster*) | *UAS-Baz dsRNA (III)* | Bloomington *Drosophila* Stock Center | RRID:BDSC_35002 | |
| Gene (*Drosophila melanogaster*) | *UAS-Notch dsRNA (III)* | Bloomington *Drosophila* Stock Center | FLYB: FBgn0004647; RRID:BDSC_28981 | |

*Continued on next page*

*Continued*

| Reagent type (species) or resource | Designation | Source or reference | Identifiers | Additional information |
|---|---|---|---|---|
| Gene (*Drosophila melanogaster*) | *UAS-Dl dsRNA (III)* | Bloomington *Drosophila* Stock Center | FLYB: FBgn0000463; RRID:BDSC_28032 | |
| Gene (*Drosophila melanogaster*) | *UAS-Nslmb-vhhGFP4 (II)* | Bloomington *Drosophila* Stock Center | RRID:BDSC_38422 | |
| Gene (*Drosophila melanogaster*) | *UAS-Nslmb-vhhGFP4 (III)* | Bloomington *Drosophila* Stock Center | RRID:BDSC_38421 | |
| Gene (*Drosophila melanogaster*) | *pnr-GAL4* | Bloomington *Drosophila* Stock Center | FLYB: FBgn0003117; RRID:BDSC_3039 | *Calleja et al., 1996* |
| Gene (*Drosophila melanogaster*) | *UAS-Nb dsRNA (III)* | Bloomington *Drosophila* Stock Center | FLYB: FBgn0002973; RRID:BDSC_35045 | |
| Gene (*Drosophila melanogaster*) | *FRT82B, neur^1F65* | *Yeh et al., 2000* | FLYB: FBgn0002932 | |
| Gene (*Drosophila melanogaster*) | *FRT82B, nls-RFP* | Bloomington *Drosophila* Stock Center | RRID:BDSC_30555 | |
| Gene (*Drosophila melanogaster*) | *hsFLP* | Bloomington *Drosophila* Stock Center | RRID:BDSC_6938 | |
| Gene (*Drosophila melanogaster*) | *FRT82B, spdo^G104* | *O'Connor-Giles and Skeath, 2003* | FLYB: FBgn0260440 | |
| Gene (*Drosophila melanogaster*) | *Spdoi-GFP (II)* | *Couturier et al., 2013* | FLYB: FBgn0260440 | BAC Rescue, Kind gift from Dr F Schweisguth |
| Gene (*Drosophila melanogaster*) | *UAS-Spdo dsRNA (II)* | Vienna *Drosophila* Resource Center | FLYB: FBgn0260440; RRID:VDRC_104092 | |
| Gene (*Drosophila melanogaster*) | *UAS-Neur dsRNA* | Bloomington *Drosophila* Stock Center | FLYB: FBgn0002932; RRID:BDSC_26023 | |
| Gene (*Drosophila melanogaster*) | *y, w, PB[y + attP-3B Neur::GFP 22A3]* | *Perez-Mockus et al., 2017* | FLYB: FBgn0002932 | BAC Rescue, Kind gift from Dr F Schweisguth |
| Antibody | Anti-Elav (rat monoclonal) | Developmental Studies Hybridoma Bank | RRID:Rat-Elav-7E8A10 | IF (1:200) |
| Antibody | Anti-cut (mouse monoclonal) | Developmental Studies Hybridoma Bank | RRID:2B10 | IF (1:500) |
| Antibody | Anti-Su(H) (goat polyclonal) | Santa Cruz | Cat# sc15813 | IF (1:500) |
| Antibody | Anti-Baz N-term (rabbit polyclonal) | *Wodarz et al., 1999* | | IF (1:1000), Kind gift from Dr A Wodarz |
| Antibody | anti-Baz-PDZ (guinea pig) | *Shahab et al., 2015* | | IF (1:1000), Kind gift from Dr A Wodarz |
| Antibody | Anti-GFP (goat polyclonal) | AbCam | Cat# Ab5450 | IF (1:500) |
| Antibody | Cy2-, Cy3- and Cy5-coupled secondary antibodies (donkey anti-goat) | Jackson Laboratories | 705-225-147, 705-165-147, and 705-175-147, respectively | IF (1:400) |
| Antibody | Cy2-, Cy3- and Cy5-coupled secondary antibodies (goat anti-rabbit) | Jackson Laboratories | 111-225-144, 111-165-144, and 111-175-144, respectively | IF (1:400) |
| Antibody | Cy2-, Cy3- and Cy5-coupled secondary antibodies (donkey anti-mouse min cross-react with rat) | Jackson Laboratories | 715-225-151, 715-165-151, and 715-175-151, respectively | IF (1:400) |

*Continued on next page*

*Continued*

| Reagent type (species) or resource | Designation | Source or reference | Identifiers | Additional information |
|---|---|---|---|---|
| Antibody | Cy2-, Cy3- and Cy5-coupled secondary antibodies (donkey anti-rat min cross-react with mouse) | Jackson Laboratories | 712-225-153, 712-165-153, and 712-175-153, respectively | IF (1:400) |
| Software, algorithm | ImageJ/FIJI | Open source Java image processing program | | https://imagej.nih.gov/ij/ |
| Software, algorithm | Countdots macro for FIJI | This study | | |
| Software, algorithm | Excel | Microsoft Office 2013 | RRID:Microsoft Excel 2013 | |
| Software, algorithm | Illustrator | Adobe Systems | RRID: Adobe Illustrator CS3 | |
| Software, algorithm | Prism | GraphPad | RRID: GrpahpadPrism | |

## Contact for reagent and resource sharing

For all kinds of requests regarding the material and methods used in this study, please refer to the lead contact, Roland Le Borgne (roland.leborgne@univ-rennes1.fr).

## Experimental model and subject details

### *Drosophila* stocks, genetics, and CRISPR-mediated homologous recombination

*Drosophila melanogaster* strains were grown and crossed at 25°C .

Somatic clones were generated using the FLP-FRT system with an hs-FLP. Crosses were passed in new tubes every 2 days and then, FLP expression was induced by at least two heat shocks (1 hr at 37 °C) from embryonic stage for *baz*$^{EH747}$ clones and from first instar larval stage for all the other clones.

The *pnr*-GAL4 driver was used to drive the expression of *UAS-Notch dsRNA; UAS-Nslmb-vhhGFP4 (II and III); UAS-Nb dsRNA; UAS-Spdo dsRNA; UAS-Neur dsRNA, UAS-Baz dsRNA, UAS-Dl dsRNA, and UAS-Numb*.

CRISPR-mediated homologous recombination was used to tag the endogenous Baz gene with His-Tag-mScarlet by inDroso (Rennes, France). The Baz-mScarlet resulted from insertion of His-Tag-mScarlet followed by an STLE linker at the amino acid position 40 of Baz (isoforms RA and RC). The gRNA was selected using the Optimal Target Finder tool (http://targetfinder.flycrispr.neuro.brown.edu/), 5' AAAGCCAAACGCAGGTGAAAAGG, cutting in the second intron (position X:17178549). Complete strategy available upon request to the lead contact.

### *Drosophila* genotypes

*Figure 1*
A *neur*-H2B-IR; Crb-GFP/Crb-GFP.
B aPKC-GFP/Y; *neur* -H2B-IR/+
C Baz-GFP/Y;; *neur*-GAP-43-IR/+
D NiGFP/Y, *neur*-H2B-RFP;; *pnr*-GAL4/+
*Figure 1—figure supplement 1*
A' Nrg-YFP/Y; neur-H2B-IR/+
B, B' Dlg-GFP/Y; ubi-Baz-mCherry/neur-H2B-IR ; neur-GAP43-IR/+
C, C' neur-H2B-IR; Crb-GFP/Crb-GFPD,
D' aPKC-GFP (*Besson et al., 2015*)/Y; neur -H2B-IR/+/Y; *neur* -H2B-IR/+
F Baz-GFP/Y;; *neur*-GAP-43-IR/+
G NiGFP/Y; *neur* -H2B-IR/+
*Figure 2*
A', B' NiGFP, FRT19A/*ubi*-mRFP-nls, w*, *hs*-FLP, FRT19A; *neur* -H2B-IR/+
B' NiGFP, FRT19A/*ubi*-mRFP-nls, w*, *hs*-FLP, FRT19A; *neur* -H2B-IR/+
C NiGFP/Y; *ubi*-Baz-mCherry/*neur*-H2B-IR;
D Baz-mScarlet/Y; E Cad-GFP/+
E-E'' NiGFP/Y; *ubi*-Baz-mCherry/*neur* -H2B-IR
*Figure 3*

A Baz-GFP/Y;; +/*pnr*-GAL4

B and E Baz-GFP/Y;; UAS-Notch RNAi/*pnr*-GAL4

C, D and F. NiGFP, Baz-mScarlet/Y; UAS-Nslmb-vhhGFP4/+; *pnr*-Gal4/*neur*-GAP43-IR

*Figure 4*

A NiGFP/Y, *neur*-H2B-RFP;; *pnr*-GAL4/+

B NiGFP/Y, *neur*-H2B-RFP;RNAi Baz/+; *pnr*-GAL4/+

C NiGFP/Y, *neur*-H2B-RFP;+/+; *pnr*-GAL4/RNAi Delta

D NiGFP/Y, *neur*-H2B-RFP;RNAi Baz /+; *pnr*-GAL4/RNAi Delta

*Figure 4—figure supplement 1*

A,A', B', C-D' and E, E' baz EH747 , FRT19A/ubi-mRFP.nls, w*, hsFLP,FRT19A;; Neur-H2B-IR/+

B, C-D' NiGFP, FRT19A/ubi-mRFP.nls, w*, hsFLP, FRT19A;; Neur-H2B-IR/+

*Figure 4—figure supplement 2*

A, A' control : NiGFP :Y, neur-H2B-RFP;+ :+; pnr-GAL4/+

B, B', E NiGFP/Y, neur-H2B-RFP; RNAi Baz/+; pnr-GAL4/+

C, C' NiGFP/Y, neur-H2B-RFP; +/+ ; pnr-GAL4/RNAi Delta

D, D', F NiGFP/Y, neur-H2B-RFP; RNAi Baz /+; pnr-GAL4/RNAi Delta

*Figure 5*

A Baz-mScarlet/Y;; attP (Bac Neur-GFP) 22A3 (*Perez-Mockus et al., 2017*) /+

B-C" control: NiGFP/Y; *neur*-H2B-IR/*hs*-FLP; FRT82B, nls-RFP/FRT82B *neur*<sup>1F65</sup>: NiGFP/Y; *neur*-H2B-IR/*hs*-FLP; FRT82B, nls RFP/FRT82B, *neur*<sup>1F65</sup>

D control: *hs*-FLP; FRT82B, nls-RFP/FRT82B *neur*<sup>1F65</sup>:*hs*-FLP; FRT82B, nls-RFP/FRT82B, *neur*<sup>1F65</sup>

D' E' NiGFP/Y; *ubi*-Baz-mCherry/Neur-H2B-IR; *pnr*-GAL4/+ or Neur RNAi

*Figure 6*

A Baz-mScarlet/Y; Numb-GFP (*Bellec et al., 2018*)/+

B-C" Ctrl: NiGFP, *neur*-H2B-RFP;; *pnr*-GAL4/+

NiGFP, *neur*-H2B-RFP;; *pnr*-GAL4/UAS-Numb RNAi

D, ENiGFP, Baz-mScarlet/Y;; *pnr*-GAL4/+

D', E NiGFP, Baz-mScarlet/Y;; *pnr*-GAL4/Numb RNAi

D", E NiGFP, Baz-mScarlet/Y;; *pnr*-GAL4/UAS-Numb

F NiGFP/Y; *ubi*-Baz-mCherry/*neur*-H2B-IR; *pnr*-GAL4/Numb RNAi

*Figure 7*

A Baz-mScarlet/Y; +/+; SpdoiGFP (*Couturier et al., 2013*)/+

B *neur*-H2B-IR/hs-FLP; FRT82B, nlsRFP/FRT82B, *spdo*<sup>G104</sup>

C, D control: NiGFP/Y; *neur*-H2B-IR/hs-FLP; FRT82B, nlsRFP/FRT82B, s*pdo* <sup>G104</sup> NiGFP/Y; *neur*-H2B-IR/hs-FLP; FRT82B, nlsRFP/FRT82B, s*pdo* <sup>G104</sup>

E, E' control: NiGFP/Y; *neur*-H2B-IR; *pnr*-GAL4/+

NiGFP/Y; *neur*-H2B-IR; *pnr*-GAL4/Spdo RNAi

NiGFP/Y; *neur*-H2B-IR; *neur*-GAP43-IR, *pnr*-GAL4/Spdo RNAi

C.NiGFP, Baz-mScarlet/Y;; *pnr*-GAL4/Spdo RNAi

D,D'. control NiGFP/Y; *ubi*-Baz-mCherry/Neur-H2B-IR; *pnr*-GAL4/+

NiGFP/Y; *ubi*-Baz-mCherry/Neur-H2B-IR; *pnr*-GAL4/Spdo RNAi

*Figure 7—figure supplement 1*

A,A', and B. NiGFP/Y; neur-H2B-IR; neur-GAP43-IR, pnr-GAL4/Spdo RNAi

C. NiGFP, Baz-mScarlet/Y ;; pnr-GAL4/Spdo RNAi

D,D'. control NiGFP/Y; ubi-Baz-mCherry/Neur-H2B-IR; pnr-GAL4/+NiGFP/Y; ubi-Baz-mCherry/Neur-H2B-IR; pnr-GAL4/Spdo RNAi

## Method details

### Immunofluorescence

Pupae aged around 17 hr after puparium formation (APF) were dissected in phosphate-buffered saline (PBS, pH 7.4) and fixed for 15 min in 4% paraformaldehyde at room temperature. They were then permeabilised performing three washes of 3 min in PBS + 0.1% Triton X-100 (PBT) and incubated with the primary antibodies (in PBT) for 2 hr at room temperature or overnight at 4 °C. After three washes of 5 min in PBT, pupae were incubated for 1 hr with the secondary antibodies (in PBT). Samples were then washed three times in PBT and once in PBS and finally mounted in 0.5% *N*-propylgallate, 90% glycerol in PBS 1× . After at least 45 min in the mounting medium, images were acquired on an LSM TCS SPE and processed using FIJI.

## Live-imaging and image analyses

Pupae aged around 16h30 APF were prepared for imaging as described previously (*Daniel et al., 2018*). Briefly, the pupa is positioned between a glass slide and a coverslip coated with a thin layer of Voltalef, the coverslip being supported anteriorly and posteriorly by columns made of four and five little coverslips. Images were acquired at 25°C on an LSM 880 AiryScan or LSM TCS SPE and processed using FIJI.

## Quantification and statistical analysis

### Statistical tests

Statistical differences between the two conditions were evaluated by an F test followed by a Student's t test using Microsoft Excel. Statistical significances were represented as follows: not significant (ns) p-value ≥ 0.05; *p-value ≤ 0.05; **p-value ≤ 0.01; ***p-value ≤ 0.001.

### Fluorescent level measurement and analysis

The NiGFP apical fluorescence level at the new pIIa/pIIb interface was measured using FIJI (version 1.52) on a z-projection summing three slices separated by 0.5 μm. A line of 30 pixels width was traced across the pIIa/pIIb interface to generate a kymograph on which another line of 20 pixels width was drawn all along the time. A plot profile then gave us the fluorescent levels (in a.u.) for each time point. These values were then corrected for the bleaching over time. To do this, on the same z-projection, we measured the fluorescent level of three different areas around the SOP, calculated the apical mean fluorescence, and determined a bleaching correction factor (t0 apical mean fluorescence/ti apical mean fluorescence) for each time point that we applied to the previous measurements at the new pIIa/ pIIb interface. Finally, we normalised to the t0 apical mean fluorescence.

### Measurement of the colocalisation rate

In order to evaluate the degree of similarity of Baz and Notch cluster dynamics, we generated kymographs from high time resolution (Δt = 2 s) acquisitions at the pIIa/pIIb new interface compared with epidermal/epidermal interfaces apically and laterally (only at pIIa/pIIb interface). We then applied the coloc two plugin from FIJI on the kymographs using the following settings: threshold regression = Costes, PSF = 4.0. We chose to use the Mander's coefficient (*Manders et al., 1993*) above auto-threshold values to evaluate the colocalisation between NiGFP and Ubi-Baz-mCherry tracks observed. Mander's coefficients represent respectively the percentage of total signal from NiGFP channel which overlaps with Ubi-Baz-mCherry signal and reciprocally the Ubi-Baz-mCherry signal which overlaps with the NiGFP signal.

## Molecular biology

To generate Neur-H2B-iRFP670 and Neur-iRFP670-GAP43 transgenic strains, we first ordered to Genewiz (Genewiz Germany Gmbh, Leipzig, Germany) a pUC57-Amp plasmids containing H2B-iRFP670 or iRFP670-GAP43 sequences flanked by StuI and SpeI restriction sites respectively on 5'-and 3'-ends. For this, we used the following sequences of H2B (*Bellec et al., 2021*), GAP43 from *Mavrakis et al., 2009*, iRFP670 (genbank KC991142) from *Shcherbakova and Verkhusha, 2013*, and pHStinger-NeurGFP from *Aerts et al., 2010*; *Barolo et al., 2000*. Details of cloning will be provided upon request.

The H2B-iRFP670 and GAP43-iRFP670 constructions were then sent to Bestgene to generate the corresponding transgenic lines with insertion at site attP40 or attP2.

## Cluster counting

To count the number of NiGFP clusters between the pIIa and pIIb nuclei, we developed a macro working with FIJI (script available upon request). Briefly: first, a threshold is applied to both pIIa and pIIb nuclei allowing for the delimitation of the nuclei inside regions of interest (ROIs). Then an ovoid mask including both nuclei ROIs is generated. From this mask, the initial nuclei ROIs are subtracted to keep only an ROI between pIIa and pIIb nuclei. Inside this ROI, the autothreshold 'RenyiEntropy' is applied and finally the clusters are detected using an 'Analyse particles'. At the end, the macro refers to the size and NiGFP fluorescence intensity of each cluster detected. Note that two erroneous situations which avoided cluster recognition by the Macro were excluded de facto from the analysis:

(a) the nuclei are too close to each other and (b) the nuclei are not positioned face to face: one is positioned above the other on the z axis. As the lateral clusters at the new pIIa/pIIb interface present characteristic size and intensity and other kinds of clusters can be detected with the NiGFP probe, we looked for a way to keep only the ones we are interested in. To do this, we observed a few samples of different genotypes and selected by eye the clusters with the right size and fluorescent intensity. We then determined size and fluorescent intensity thresholds. For size, the thresholds were constant for the different samples and we fixed the minimal cluster area at 0.03 µm$^2$ and the maximal cluster area at 0.2 µm$^2$. As for the minimal intensity threshold, we found a linear correlation with the apical mean fluorescence: intensity threshold = 0.2654 × apical mean fluorescence +227.6. We applied these two thresholds successively to the images analysed.

### Baz quantification

To quantify the intensity of Baz signal present at the apical and lateral clusters at the pIIa-pIIb interface (*Figure 6*), we sum the fluorescence signal of three apical planes at the level of AJ and between –3 and –6 µm below the apical interface, respectively. The mean fluorescence intensity within an ROI of 1 µm × 3 µm (apical interface) or 1 µm × 4 µm (lateral interface) were measured and normalised to neighbouring epidermal cell interfaces' mean fluorescence intensity.

## Acknowledgements

We thank Y Bellaïche, J Januschke, F Schweisguth, A Wodarz, the Bloomington *Drosophila* Stock Center, the Vienna *Drosophila* Resource Center, InDroso, and the Developmental Studies Hybridoma Bank for providing fly stocks and antibodies. We also thank the Microscopy Rennes Imaging Center-BIOSIT facility. We thank Emeline Daniel for the initial observations of Baz localisation, Nathalie Bécot for her help in fly genetics, Thomas Esmangart de Bournonville and Antoine Guichet for critical reading of the manuscript. This work was supported in part by the ARC (post doctoral fellowship), La Ligue contre le Cancer-Equipe Labellisée (RLB), and the Association Nationale de la Recherche et de la Technologie programme PRC Vie, santé et bien-être CytoSIGN (ANR-16-CE13-004-01 to RLB).

## Additional information

### Funding

| Funder | Grant reference number | Author |
| --- | --- | --- |
| Agence Nationale de la Recherche | ANR-16-CE13-004-01 | Roland Le Borgne |
| Ligue Contre le Cancer | Equipe Labellisée | Roland Le Borgne |

The funders had no role in study design, data collection and interpretation, or the decision to submit the work for publication.

### Author contributions

Elise Houssin, Conceptualization, Formal analysis, Investigation, Methodology, Resources, Validation, Visualization, Writing - original draft; Mathieu Pinot, Formal analysis, Investigation, Methodology, Validation, Visualization, Writing - original draft; Karen Bellec, Investigation, Methodology, Resources, Visualization; Roland Le Borgne, Conceptualization, Formal analysis, Funding acquisition, Investigation, Supervision, Validation, Visualization, Writing - original draft, Writing - review and editing

### Author ORCIDs

Mathieu Pinot http://orcid.org/0000-0003-4029-3437
Karen Bellec http://orcid.org/0000-0001-5321-3921
Roland Le Borgne http://orcid.org/0000-0001-6892-278X

### Ethics

All expereiments were done according to ethics rules concerning the use of Drosophila melanogaster, and GMOs declared in the declaration of contained use of genetically modified organisms (GMOs) of

containment class 1 n °2898 from the Ministère de l'Enseignement Supérieur, de la Recherrche et de l'Innovation to R.L.B.

## Decision letter and Author response

Decision letter https://doi.org/10.7554/eLife.66659.sa1
Author response https://doi.org/10.7554/eLife.66659.sa2

## Additional files

### Supplementary files
• Transparent reporting form

### Data availability
All data generated or analysed during this study are included in the manuscript and supporting files. Source data files will be provided for each of the Figures.

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
