## [Decision Letter]

**Acceptance summary:**

This study reveals the existence of a novel Notch receptor containing signalling hub, organised by Sanpodo and Par3, which operates in cell fate decisions in the peripheral nervous system of *Drosophila*. These Notch clusters are modulated by components of the Notch signaling pathway, and are proposed to reinforce Notch signaling by concentrating ligands and receptors. These findings are highly relevant to different areas of biology including membrane biology, cytokinesis, PAR polarity, Notch signaling and cell fate decisions making.

**Decision letter after peer review:**

Thank you for submitting your article "Par3 cooperates with Sanpodo for the assembly of Notch signaling clusters following asymmetric division in *Drosophila*" for consideration by *eLife*. Your article has been reviewed by 3 peer reviewers, one of whom is a member of our Board of Reviewing Editors, and the evaluation has been overseen by K VijayRaghavan as the Senior Editor. The reviewers have opted to remain anonymous.

Essential revisions:

1. The results mentioned in the discussion, but not shown (baz δ double mutants resulting in strong Notch lof phenotype) should be a way to improve the functional relevance of the findings.

2. The authors show that in the absence of Baz the Notch clusters are decreased in intensity, but this does only slightly affect Notch signaling. This issue should be discussed more carefully, and the word "signalling" should be removed from the title.

3. For other revisions, see individual recommendations for the authors below.

*Reviewer #1 (Recommendations for the authors):*

– This manuscript addresses two important questions: the redistribution of polarity/junctional proteins during cytokinesis of the SOP on the one hand and Notch signaling on the other hand. Although these two aspects are certainly linked, intermingling the experimental approaches to address these questions sometimes confuses the reader and fails short in an in-depth analysis of each individual aspect (in fact, the authors seem to have several unpublished data supporting each of these aspects). For example, the somewhat lengthy discussion on a possible role of Crb in Notch signaling at the pIIa/pIIb interface does not really help (it lacks other references on the Notch-Crb interactions).

– Line 184, 552: Is there really a "remodeling of apico-basal cell polarity" taking place? There are changes in the distribution of some polarity markers, in particular at the pIIa/pIIb interface, which is clearly different from that of a dividing epidermal cell, but does this justify the conclusion that apico-basal polarity is remodeled?

– Page 22: the role of Numb in repressing Notch has been shown previously (line 444 and 454). Do "increased transient signals of Notch" really support a Notch "gain-of-function phenotype" and hence more Notch signaling (line 444)? It rather correlates with this. And does loss of Numb only affect the lateral Baz clusters or also the apical ones?

*Reviewer #2 (Recommendations for the authors):*

I feel the results mentioned in the discussion, but not shown (baz δ double mutants resulting in strong Notch lof phenotype) could be a way to improve the functional relevance of the findings.

First paragraph of the results is important to shape the argument of a specialised membrane. However, I find it displaced in the manuscript, it defocusses. Maybe that can be placed more elegantly? Starting with the intriguing observation that polarity seems to be remodelled after anaphase onset and the Baz clusters seems better.

A coherent definition of the relevant cell-cell contact sites containing the here described BAZ clusters could help. Maybe define the relevant pool described here as "lateral interface pool" and use throughout? I assume this is related to the NICD pool basal to the midbody described before? Please consistently label the surfaces in all figures with a box or arrow heads.

Kymographs: they are used to measure co-localization, which is good, could they also be used to make a statement about the ability of the clusters to diffuse laterally? The "snap button" concept somewhat suggests (at least to me) some form of spatial stability and limited lateral mobility. Are these stable or rather immobile clusters? If so is this mobility changed in the mutant situations analysed?

Quantification of clusters in neuralised mutants (Figure 5C) cannot be understood as is.

The baz mutant analysis is interesting, but does not necessarily assign a function to the identified clusters.

I am not sure the paragraph line 577 is directly relevant to the findings specifically reported here.

Line 613. The idea if this paragraph is nice. It could benefit from a graphical abstract type summary incorporating the findings in form of a working model.

Line 217, I can't follow how epidermal cell division was traced.

Line 290, given that the accumulation of NICD is only transient in neur mutants, the conclusion can only be ,…but fail to be disassembled in a timely manner.

Figure 1, rather than the current figures D' and D' a little schematic to orient the reader could help. The reference to the difference in NICD levels in the SOP daughter cell nuclei is a bit out of place here.

A coherent definition of the relevant cell-cell contact sites containing the here described BAZ clusters could help. Maybe define the relevant pool described here as "lateral interface Baz pool" or similar?

Numb RNAi, effect on NICD cluster presence is transient, is that true for Baz? Not clear.

S980 on Baz is a conserved aPKC phosphorylation site, not Par-1, at least not that I am aware of.

Figure legends could be improved for clarity.

Figure 1 arrowheads in A and B not annotated. Dashed lines an A and B not annotated

Figure 1D not mentioned in the text.

Figure 2, quantification of the clonal analysis of clusters is missing (sample size, repeats?).

Figure 2 E', something is not right here. The annotation misses apical and lateral, I think. Maybe better represent using confidence intervals? The kymographs for apical and lateral in SOPs look somewhat different, yet the coefficient is almost the same.

Figure 4. I assume this is a standard test in the field, but nonetheless the markers Elav, Cut, Su(H) should be shown in a control. Figure 4 lacks quantification of Baz lateral clusters to conclude they are Notch independent.

Figure 5A anterior localization of Neur is not clear. Figure legend and main text diverge here a bit.

Figure 5C, colour code in quantification unclear.

Figure 5D, Cut? Wrong label, I guess, whether fixed or live analysis unclear.

Figure 5D is not quantified.

Figure 5 E should be quantified using the colocalization coefficient shown before.

Figure 6 C-C'colour code unclear/not annotated.

Figure 6F overexpression effect of Numb and Baz not quantified.

*Reviewer #3 (Recommendations for the authors):*

1. To facilitate understanding the SOP/pIIa/pIIb cellular system, it would be extremely helpful to add a drawing of the SOP and its daughter cells surrounded by pupal epidermal cells in Figure 1.

2. In the manuscript, a large number of transgenes encoding fluorescent proteins are used, but some have not been sufficiently explained to non-expert readers. For instance, it shoud be explained why the neur::H2B-IRF670 transgene was used for identification of the SOP and its daughter cells and which structures are marked by this transgene. The same applies to the GAP43-IR transgene and several others.

3. Figure 4A and B are supposed to show the excess of SOP specification upon loss of N function in the pupal notum. Here, a wild type control is missing. Furthermore, the image quality of Figure 4A is insufficient (large pixels) and it is not clear to me what the image is actually showing. The scale bars for all panels in Figure 4 are given as 5 µm, but I am quite sure that this is incorrect and should be checked by the authors.

4. Page 10, top paragraph: The authors talk here about data that they do not show. The data either need to be shown or the statement should be deleted. This also applies to several other text passages in the manuscript.

5. Figure 1: The labeling of pIIa and pIIb in the panels is not entirely clear. It would be better to add a line pointing to the center of the respective cells. This also applies to many other figures in the manuscript.

6. Figure 2: The labeling of panels A' and B' is incorrect. It should be nls-RFP instead of nls-GFP.

7. Figure 3C, C': Can the authors make any statement about the statistical significance of the differences between control and baz mutant daughter cell interfaces? The same question applies to Figure 5C', C'; Figure 6C', C'; Figure S2C, C' and Figure S3C, C'.

8. Legend to Figure S2: In the first sentence it is stated that the green Baz staining can be seen in B and B'. It should be A and A'.

9. Figure 4D, E: It is not clear to me what is labeled by GAP43-IR. Are the white structures supposed to be the membranes of pIIa and pIIb cells? These cells should be labeled explicitly in the figure.

10. Figure 5D: In the labeling of this panel, both Baz and Cut are in green, which is probably incorrect.

11. In general, the text needs some thorough editing with respect to English grammar.

---

## [Author Response]

Essential revisions:1. The results mentioned in the discussion, but not shown (baz δ double mutants resulting in strong Notch lof phenotype) should be a way to improve the functional relevance of the findings.

In the first version of the manuscript, we reported that the single loss of Delta or Baz were giving a partially penetrant Notch *loss-of-function* phenotype, a tufting and a moderate neurogenic phenotype, respectively. We proposed that loss of Baz generated a sensitised background for Notch signalling.

Based on reviewer suggestions, we have now included the analyses of adult phenotypes and bristle lineages to show the changes in cell fate acquisition upon single versus simultaneous loss of the Notch ligand and Baz (Figure 4—figure supplement 2 A-D’). We report a highly penetrant phenotype in the double RNAi situation, a phenotype reminiscent to loss of Notch activity, with a bald cuticle and absence of Su(H) positive cells. In addition, we have performed live imaging of Notch dynamics and quantify the NiGFP signal present in apical and lateral interface clusters upon silencing of Baz and Delta (these new data are presented in Figure 4). We now provide evidence that the NiGFP clusters that failed to dissociate in a timely manner upon single depletion of Delta are no longer detected upon concomitant loss of Baz and Delta, supporting the notion of sensitised background.

These data are presented in Figure 4 and Figure 4—figure supplement 2.

2. The authors show that in the absence of Baz the Notch clusters are decreased in intensity, but this does only slightly affect Notch signaling. This issue should be discussed more carefully, and the word "signalling" should be removed from the title.

For the revision, we have repeated the Baz RNAi experiments and show that, despite the fact that silencing of Baz results in a total loss of Baz immunostaining (Figure S3 E-F), i.e. efficient depletion of Baz, Notch signalling was only mildly impacted (Figure S2E, E’ and S3 C, C’). Based on the result of the simultaneous depletion of Baz and Delta, we propose three alternative, non-mutually exclusive models for Baz activity:

1 – Baz oligomerisation triggers the assembly of the Baz clusters independently of Notch/Spdo clusters (the ones detected upon Baz silencing). By analogy with the formation of Spot adherens junctions, we hypothesise that Baz clusters and Notch/Spdo clusters interact with each other to assemble the Baz/Notch/Spdo clusters.

2 – Baz, by its virtue of interacting with phosphoinositides (PIP3) and oligomerise, could serve as an exocyst receptor at interface membrane to promote local delivery and localisation of Notch/Spdo. This polarised Baz/Exocyst targeting (inherited from polarity remodelling during SOP cytokinesis) would account for the preferential targeting and cluster assembly at the pIIa-pIIb interface rather than evenly distributing on the entire plasma membrane of the pIIa and pIIb cells.

In both models 1 and 2, Notch/Spdo/Baz clustering by increasing the local concentration of Notch receptor would favour the transactivation by the ligand Delta present in the pIIb cell.

3 – Baz and Ser. Silencing of Delta recapitulates the complete loss of Delta reported by the Jan’s lab (Zeng et al., 1998). The resulting tufting phenotype indicates that Delta is strictly required during lateral inhibition, to enable the SOP selection, and that Serrate, the second Notch ligand, cannot substitute for Delta. Of note, Notch-dependent binary fate acquisition takes place normally upon loss of Delta. The authors propose that Serrate activity fully substitutes for that of Delta during asymmetric cell division. Thus, only the double *Delta*, *Serrate* mutant mimics loss of Notch. The fact that double silencing of Baz and Delta phenocopies loss of Notch led us to propose the following third possibility, according to which Baz regulates Serrate activities.

These three possibilities are now discussed in the revised manuscript, and future studies will aim at deciphering the exact molecular mechanism by which Baz controls clusters assembly and Notch signalling. To follow the referees’ recommendation, we have removed the word ‘signalling’ from the title.

3. For other revisions, see individual recommendations for the authors below.Reviewer #1 (Recommendations for the authors):– This manuscript addresses two important questions: the redistribution of polarity/junctional proteins during cytokinesis of the SOP on the one hand and Notch signaling on the other hand. Although these two aspects are certainly linked, intermingling the experimental approaches to address these questions sometimes confuses the reader and fails short in an in-depth analysis of each individual aspect (in fact, the authors seem to have several unpublished data supporting each of these aspects). For example, the somewhat lengthy discussion on a possible role of Crb in Notch signaling at the pIIa/pIIb interface does not really help (it lacks other references on the Notch-Crb interactions).– Line 184, 552: Is there really a "remodeling of apico-basal cell polarity" taking place? There are changes in the distribution of some polarity markers, in particular at the pIIa/pIIb interface, which is clearly different from that of a dividing epidermal cell, but does this justify the conclusion that apico-basal polarity is remodeled?

In the revised manuscript, we have avoided the notion of polarity remodelling to restrict the description to the polarity markers we analysed, without prejudging the overall polarity of the pIIa-pIIb interface. We have also removed the section on crumbs from the discussion, as in its current form it is a proposal that is lacking experimental support.

– Page 22: the role of Numb in repressing Notch has been shown previously (line 444 and 454). Do "increased transient signals of Notch" really support a Notch "gain-of-function phenotype" and hence more Notch signaling (line 444)? It rather correlates with this.

We agree with the reviewer that this was a correlation. We have amended the text as follows:

“Inhibition of Numb that results in a *gain-of-function* Notch phenotype (Guo et al., 1996) is accompanied by increased Notch transient signals at the apical pIIa-pIIa-like interface (Figure 6B-C), and a higher number of brighter clusters at the lateral interface of NiGFP”.

In the course of the revision, we have performed preliminary photoconversions and photo-tracking of Nimaple3 present at the apical interface of the SOP daughters as we did in (Bellec et al., 2021) in Numb RNAi context. We observed that the Notch present at the interface upon Numb silencing at t15min is present in both SOP daughter nuclei at t35 min (data not shown). Although these data support the idea that ‘increased Notch transient signals’ are responsible for the *Notch* ‘*gain-of-function* phenotype’, these data are qualitative and not quantitative (i.e. they do not answer the question: ‘Is it the unique source of active Notch?’), and therefore were not included in the revised manuscript.

And does loss of Numb only affect the lateral Baz clusters or also the apical ones?

Although silencing of Numb results in a higher and longer lasting apical NiGFP signal (revised Figure 6 B,C), the number of apical Baz signals measured at t = 14 min after the onset of anaphase is not significantly different than in control (revised Figure 6 D, E). Unfortunately, the rapid photobleaching of the (Baz-)Scarlet signal prevented us from performing a kinetic analysis, so we cannot determine if the Baz signal persists longer after Numb depletion.

Reviewer #2 (Recommendations for the authors):I feel the results mentioned in the discussion, but not shown (baz δ double mutants resulting in strong Notch lof phenotype) could be a way to improve the functional relevance of the findings.

This point has been addressed in the revised manuscript, see our response to Reviewer 1, above.

First paragraph of the results is important to shape the argument of a specialised membrane. However, I find it displaced in the manuscript, it defocusses. Maybe that can be placed more elegantly? Starting with the intriguing observation that polarity seems to be remodelled after anaphase onset and the Baz clusters seems better.

We do not agree with the reviewer. We kept the order as in the initial version, but we simplified the message and added schematic representations of the cells and marker distribution.

A coherent definition of the relevant cell-cell contact sites containing the here described BAZ clusters could help. Maybe define the relevant pool described here as "lateral interface pool" and use throughout? I assume this is related to the NICD pool basal to the midbody described before? Please consistently label the surfaces in all figures with a box or arrow heads.

We thank the reviewer for this suggestion. In the revised text, we have systematically referred to ‘apical and lateral interface clusters’. Lateral clusters are indeed the ones located basally to the midbody.

In the revised figures we have systematically labelled the pIIb and pIIa cells with red and blue dots, respectively, and lateral clusters with arrowheads.

Kymographs: they are used to measure co-localization, which is good, could they also be used to make a statement about the ability of the clusters to diffuse laterally?

We did not comment on the ability of the cluster to laterally diffuse as we have no proof/argument for lateral diffusion versus displacement of the entire lateral interface that would not be restricted to clusters themselves.

Preliminary FRAP analyses suggest that NiGFP and Baz clusters are constantly replenished about a minute (data not shown). However, based on the dynamics of the clusters and the time required to define the ROI to FRAP, run the FRAP, and time of imaging, these experiments are extremely difficult to perform and interpret. Although this is the type of experiment we would have liked to implement in the revised manuscript (control and mutant situation), it is still too challenging technically.

The "snap button" concept somewhat suggests (at least to me) some form of spatial stability and limited lateral mobility. Are these stable or rather immobile clusters?

To our mind, the snap button does not preclude stability. We named them ‘snap button’ because we can follow the traces on kymographs at least over 5 min (‘apparently stable over 5 min’). In addition, the ½ time recovery of Notch in clusters is in the minute long, arguing for a replenishment of NiGFP in the cluster (see comment above). Because we can follow the track without apparent interruption on the kymograph, we favour the idea that the clusters are the same structures, able to be displaced laterally and replenished, although we cannot firmly exclude that what we refer to as lateral displacement is assembly of a new cluster next to a previous one that has just been consumed. An even higher spatio-temporal resolution would be needed to discriminate between these two possibilities.

If so is this mobility changed in the mutant situations analysed?

We did not notice obvious changes in the lateral displacement of the clusters in kymographs between the different experimental situations analysed. As indicated above, the FRAP experiment (if doable) would have been informative.

Quantification of clusters in neuralised mutants (Figure 5C) cannot be understood as is.

In the revised version, we modified the representation of the number and intensities of cluster.

The baz mutant analysis is interesting, but does not necessarily assign a function to the identified clusters.

See our response to the biological relevance of the Baz Notch clusters, addressing the main concern of the three reviewers.

I am not sure the paragraph line 577 is directly relevant to the findings specifically reported here.

We agree with the reviewer, and we removed the section concerning Crumbs in the discussion of the revised manuscript.

Line 613. The idea if this paragraph is nice. It could benefit from a graphical abstract type summary incorporating the findings in form of a working model.

We have now added schematical representations of the distribution corresponding markers along the nascent pIIa-pIIb interface in Figures 1 to 7.

Line 217, I can't follow how epidermal cell division was traced.

The revised text (first paragraph of the Results section) has been amended as follows:

‘The progression of mitosis was tracked by cell shape. Cells in prometaphase are spherical, and the metaphase to anaphase transition is determined by the moment when the sphericity of the cells is lost, prior to adoption of a peanut shape. In every case, the transition from metaphase to anaphase was considered to be t0 (time is indicated in min:sec).’

Line 290, given that the accumulation of NICD is only transient in neur mutants, the conclusion can only be ,…but fail to be disassembled in a timely manner.

We thank the reviewer for pointing out this error. We have corrected the text accordingly.

Figure 1, rather than the current figures D' and D' a little schematic to orient the reader could help. The reference to the difference in NICD levels in the SOP daughter cell nuclei is a bit out of place here.

We agree with the reviewer, and have removed the panel’s D' and D" signs. We have also added schematic representations of the locations of the different markers used.

A coherent definition of the relevant cell-cell contact sites containing the here described BAZ clusters could help. Maybe define the relevant pool described here as "lateral interface Baz pool" or similar?

We agree with the reviewer, and have amended the text accordingly.

Numb RNAi, effect on NICD cluster presence is transient, is that true for Baz? Not clear.

The NiGFP clusters are indeed transient upon Numb silencing. In contrast, Baz lateral clusters are still present after NiGFP signal is no longer detectable. This is also the case in the wild type situation. The reason for this longer persistence time remains unknown at present. Analysis of these cluster dynamics with FRAP could bring some information (time residence). Unfortunately, the rapid photobleaching of the (Baz-)Scarlet signal prevented us from performing a kinetic analysis, so we cannot determine if the Baz clusters’ signal upon Numb depletion differs from the control situation.

S980 on Baz is a conserved aPKC phosphorylation site, not Par-1, at least not that I am aware of.

We agree and thank the reviewer. This part has been removed. As we did not show the data in the current manuscript, we have removed this part of the text (see Reviewer 3, point 4).

Figure legends could be improved for clarity.

We have taken over and amended each of the legends of the figures to simplify them, to describe the content only without bringing any conclusion/interpretation.

Figure 1 arrowheads in A and B not annotated. Dashed lines an A and B not annotatedFigure 1D not mentioned in the text.

We have now added the missing information (arrowheads, dashed lines….) in the figure legends, and Figure 1D is quoted in the revised manuscript.

Figure 2, quantification of the clonal analysis of clusters is missing (sample size, repeats?).

We apologise for this oversight. We have now added the quantification (n=8, low number because finding the right arrangement of NiGFP+ and NiGFP- SOPs is not very frequent) of the three independent clonal analyses performed in Figure 2A.

Figure 2 E', something is not right here. The annotation misses apical and lateral, I think. Maybe better represent using confidence intervals? The kymographs for apical and lateral in SOPs look somewhat different, yet the coefficient is almost the same.

We verified and corrected the annotation and explain the way the Mander’s coefficient were obtained in the Methods section ‘Measurement of the colocalisation rate’. For epidermal cells we quantified the apical pool (the only one detectable) and compared it to NiGFP signal present at the apical and lateral interface of SOP daughters.

Figure 4. I assume this is a standard test in the field, but nonetheless the markers Elav, Cut, Su(H) should be shown in a control.

We have now added a control lineage in the revised manuscript (Figure 3A and S3A).

Figure 4 lacks quantification of Baz lateral clusters to conclude they are Notch independent.

The quantitation has been made and is now included in the revised Figure 3 E, F’. The number of lateral Baz Clusters is not affected by the loss of Notch (silencing and degradFP).

Figure 5A anterior localization of Neur is not clear. Figure legend and main text diverge here a bit.

In revised Figure 5A, we have replaced the panel of the basal plane by a -4µm to -5µm maximal projection (three confocal sections) to better show the anterior enrichment of Neuralized (in the original figure, a single confocal plane was shown). We have modified the figure legends according to the overall reviewer comments. The ‘divergence’ between the legend and the text has been fixed.

Figure 5C, colour code in quantification unclear.

We have now modified the text accordingly and specified the colour code for each genotype in the figures and in the legends.

Figure 5D, Cut? Wrong label, I guess, whether fixed or live analysis unclear.

We apologise for the lack of clarity. This panel corresponds to a fixed specimen and Baz and Cut were both detected with secondary antibodies coupled to Cy2 (same channel for image analyses).

Figure 5D is not quantified.

In the original version, we decided not to quantify the number of Baz lateral clusters on images of fixed specimen as we cannot ascertain the timing of the pIIa-pIIb stage analysed relative to the anaphase onset (the to). In the revised version, we have quantitated the number of Baz lateral clusters (ubi-Baz-mCherry) at t10 in control versus Neur RNAi condition. This new dataset showing the increased number of Baz lateral clusters upon Neur silencing is presented in Figure 5 D’,D’’.

Figure 5 E should be quantified using the colocalization coefficient shown before.

We thank the reviewer for this suggestion that we have followed and present in Figure 5E’. The extent of colocalisation between NiGFP and Baz-mCherry is shown to be independent of Neur function.

Figure 6 C-C'colour code unclear/not annotated.

As stated above, this has now been fixed.

Figure 6F overexpression effect of Numb and Baz not quantified.

The effect of Numb overexpression or depletion on the number and fluorescence intensity of Baz lateral clusters have now been quantified and are presented in Revised Figure 6 E.

Reviewer #3 (Recommendations for the authors):1. To facilitate understanding the SOP/pIIa/pIIb cellular system, it would be extremely helpful to add a drawing of the SOP and its daughter cells surrounded by pupal epidermal cells in Figure 1.

We thank the reviewer for this remark. We have now included schematic representation (revised Figures 1, 4, 5, and 6) that should help the readers understand the localisations in 3D.

2. In the manuscript, a large number of transgenes encoding fluorescent proteins are used, but some have not been sufficiently explained to non-expert readers. For instance, it shoud be explained why the neur::H2B-IRF670 transgene was used for identification of the SOP and its daughter cells and which structures are marked by this transgene. The same applies to the GAP43-IR transgene and several others.

We have now added the following information in the first paragraph of the revised manuscript:

“SOPs and daughters were identified using the nuclear marker Histone2B-IRFP (H2B-IR) or the plasma membrane marker GAP43-IRFP, expressed under the SOP specific neur minimal promoter. GAP43 consist in the first 20 amino acids of growth-associated protein 43 (GAP43), which contain a dual palmitoylation signal that tightly anchors the protein to the inner leaflet of the PM.”

3. Figure 4A and B are supposed to show the excess of SOP specification upon loss of N function in the pupal notum. Here, a wild type control is missing.

We apologise for this oversight. Control situation has now been included in revised Figure 3A (previous Figure 4), as well as that of Baz and Delta RNAi (Figure S3A,A’)

Furthermore, the image quality of Figure 4A is insufficient (large pixels) and it is not clear to me what the image is actually showing.

The quality of the data presented figure 4A is of sufficient quality to appreciate the excess of Neurons (Elav+ cells in blue) at the expense of socket cells (Su(H) + cells in magenta). The residual, low-intensity signal in the magenta channel is background.

The scale bars for all panels in Figure 4 are given as 5 µm, but I am quite sure that this is incorrect and should be checked by the authors.

The scale bars were missing in Figure 4 D, E, we thank the reviewer for pointing out this problem.

4. Page 10, top paragraph: The authors talk here about data that they do not show. The data either need to be shown or the statement should be deleted. This also applies to several other text passages in the manuscript.

We have modified the text in accordance and deleted the statement where the data was not shown.

5. Figure 1: The labeling of pIIa and pIIb in the panels is not entirely clear. It would be better to add a line pointing to the center of the respective cells. This also applies to many other figures in the manuscript.

To facilitate the identification of the SOP daughter cells, in the revised version we have put a blue and red dot to colour-code the pIIa (and pIIa-like) and pIIb (and pIIb-like) cells respectively *.*

6. Figure 2: The labeling of panels A' and B' is incorrect. It should be nls-RFP instead of nls-GFP.

We thank the reviewer for pointing out this mistake. The text has now been modified.

7. Figure 3C, C': Can the authors make any statement about the statistical significance of the differences between control and baz mutant daughter cell interfaces? The same question applies to Figure 5C', C'; Figure 6C', C'; Figure S2C, C' and Figure S3C, C'.

We have changed the mode of representation and instead of making categories of number of clusters and different fluorescent intensities, we have now illustrated the number of lateral interface clusters/cell/condition (panels Figure 4E ) and their individual fluorescence intensities (panels Figure 4E) in revised Figures 4, 5 and 6, and have also applied statistical tests.

8. Legend to Figure S2: In the first sentence it is stated that the green Baz staining can be seen in B and B'. It should be A and A'.

We thank the reviewer for pointing out this mistake, which we have corrected in revised Figure 4—figure supplement 1.

9. Figure 4D, E: It is not clear to me what is labeled by GAP43-IR. Are the white structures supposed to be the membranes of pIIa and pIIb cells? These cells should be labeled explicitly in the figure.

As mentioned above and in the revised text, GAP43 consists in the first 20 amino acids of growth-associated protein 43, which contain a palmitoylation signal that anchors it to the plasma membrane. It is fused to the IRFP670 and expressed under a SOP specific promoter, so the plasma membrane of every SO and daughters are labelled with it. In panels D and E, the reduction of Notch activity give rise to an excess of SOP specification, explaining the density of GAP43 cells, especially in 4D. The SO daughters, GAP43+ cells are identified by the fheir timing of division (around 16h00 APF), the planar orientation of the cell division, and the initial asymmetry of daughter cell sizes. In the revised Figure 3 (previous Figure 4) they are identified by coloured dots.

10. Figure 5D: In the labeling of this panel, both Baz and Cut are in green, which is probably incorrect.

No, it is correct, they are in the same channel (see above, response to Reviewer 2)

11. In general, the text needs some thorough editing with respect to English grammar.

We have carefully edited the text and had it corrected by the specialist company, Scribbr (https://www.scribbr.fr/).